

# Strings and membranes from $\mathcal{A}$-theory five brane

**Machiko Hatsuda[1,2]⋆, Ondřej Hulík[3], William D. Linch[4],**
**Warren D. Siegel[5,6], Di Wang[6] and Yu-Ping Wang[6]**

**1** Department of Radiological Technology, Faculty of Health Science,
Juntendo University, Yushima, Bunkyou-ku, Tokyo 113-0034, Japan
**2** KEK Theory Center, High Energy Accelerator Research Organization,
Tsukuba, Ibaraki 305-0801, Japan
**3** Institute for Mathematics Ruprecht-Karls-Universitat Heidelberg,
69120 Heidelberg, Germany
**4** Thomas Jefferson High School for Science and Technology, Alexandria, VA 22312, USA
**5** C. N. Yang Institute for Theoretical Physics. Stony Brook, NY 11794, USA
**6** Department of Physics, SUNY Stony Brook University, Stony Brook, NY 11794, USA

⋆ mhatsuda@juntendo.ac.jp

## Abstract

The $\mathcal{A}$-theory takes U-duality symmetry as a guiding principle, with the SL(5) U-duality symmetry being described as the world-volume theory of a 5-brane. Furthermore, by unifying the 6-dimensional world-volume Lorentz symmetry with the SL(5) spacetime symmetry, it extends to SL(6) U-duality symmetry. The SL(5) spacetime vielbein fields and the 5-brane world-volume vielbein fields are mixed under the SL(6) U-duality transformation. We demonstrate that consistent sectionings of the SL(6) $\mathcal{A}$5-brane world-volume Lagrangian yield Lagrangians of the $\mathcal{T}$-string with O(D,D) T-duality symmetry, the conventional string, the $\mathcal{M}$5-brane with GL(4) duality symmetry, and the non-perturbative M2-brane in supergravity theory. The GL(4) covariant Lagrangian of the $\mathcal{M}$5-brane derived in this manner is a new, perturbatively quantizable theory.

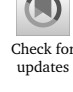

# 1  Introduction

## 1.1  An overview of $\mathcal{A}$-theory formalism

Superstring theory is regarded as a strong candidate for a unified theory that encompasses all four fundamental interactions. However, rather than a single theory, it currently exists in the form of six distinct theories: five superstring theories and M-theory. These six theories are intricately connected via S-duality and T-duality, forming a hexagonal network of dualities. Each of these six theories is defined on its own characteristic brane, and together they provide complementary descriptions of the broader structure of superstring theory. A key open question is what kind of theoretical framework can provide a unified formulation of these six theories.

S-duality and T-duality are unified as U-duality, whose structure reflects the group-theoretic inclusiveness of these dualities. The symmetry group of a theory with manifest S-duality is given by GL(D+1) which extends the spacetime diffeomorphism group GL(D) by incorporating the S-duality group SL(2) as a subgroup. GL(D+1) and the T-duality symmetry O(D,D) are embedded within the exceptional group $E_{D+1}$ which serves as the symmetry group of U-duality. U-duality is therefore expected to relate all six superstring theories in a coherent manner. A theory that explicitly manifests this U-duality symmetry is referred to as "$\mathcal{A}$-theory", and major objective is to construct a perturbative formulation of this theory that facilitates a systematic analysis of its quantum aspects (see [1] for review).

In 1995, Witten proposed M-theory as a strong coupling limit of the type IIA superstring theory via S-duality [2]. The low-energy effective theory of M-theory is 11-dimensional supergravity, whose diffeomorphism symmetry is GL(11), combining the 10-dimensional diffeomorphism and the SL(2) S-duality symmetry. We refer to a world-volume theories that exhibit manifest GL(D+1) symmetry as $\mathcal{M}$-theory. In 1996, Vafa proposed F-theory as a framework related to the type IIB superstring theory via S-duality [3]. For further discussion, see, for example [4].

In 1993, Siegel proposed a string theory guided by the T-duality symmetry O(D,D) [5–7]. This framework employs the idea of doubling spacetime coordinates, where the 2D-dimensional spacetime coordinates are treated as vectors in the O(D,D) representation [8]. The geometry generated by the O(D,D) covariant current algebra is the stringy gravity theory with $B$-field, where gauge fields are parameters of the coset O(D,D) over the doubled Lorentz group. We refer to this as $\mathcal{T}$-theory and the T-duality covariant string as $\mathcal{T}$-string. Later Hull, Zwiebach, and Hohm proposed a theory of O(D,D) covariant background fields, known as Double Field Theory (DFT) [9,10]. This theory serves as the low-energy effective gravitational description of $\mathcal{T}$-string theory. In order to consistently reduce the doubled 2D-dimensional spacetime to the physical D-dimensional spacetime, the section condition was employed, which corresponds to the zero-mode of the Virasoro constraint $\mathcal{S}=0$. For detailed reviews, see [11,12]. The $\mathcal{T}$- theory was extended to incorporate $\mathcal{N}=2$ supersymmetry including Ramond–Ramond gauge fields based on the doubled non-degenerate super-Poincaré group [13–17]. All bosonic component fields represent supersymmetrized O(D,D) while fermions represent only O(D-1,1)$^2$. Incorporating S-duality requires extending the symmetry to the exceptional U-duality group. Consequently, the exceptional group $\mathrm{E}_{D+1}$ is expected to be related to a subgroup of the doubled non-degenerate super-Poincaré group.

Siegel, Linch and Polacek subsequently proposed brane theories guided by the U-duality symmetry ( $\mathrm{E}_{D+1}$ ) [13, 18, 19]. Since the representation of exceptional groups varies with dimension, the theory is labeled by the spacetime dimension $D$ when reduced to string theory. This is called D= $D$ $\mathcal{A}$-theory. According to the classification of Lie algebras, removing a specific node in the Dynkin diagram of $\mathrm{E}_{D+1}$ reduces it to the GL(D+1) Dynkin diagram. This corresponds to reducing the spacetime dimensions of $\mathcal{A}$-theory using the Virasoro constraint, recovering the aforementioned $\mathcal{M}$-theory. Conversely, removing another node in the $\mathrm{E}_{D+1}$ Dynkin diagram reduces it to the O(D,D) Dynkin diagram. This corresponds to reducing the world-volume dimensions of $\mathcal{A}$-theory using the Gauß law constraint $\mathcal{U}=0$, recovering the aforementioned $\mathcal{T}$-theory. As a consequence of these dual reductions, $\mathcal{A}$-theory is consistently described by branes covariant under the exceptional group. Both the spacetime coordinates and world-volume coordinates of the branes are representations of the exceptional group, ensuring that the brane current algebra is covariant under the exceptional group. Moreover, a perturbative Lagrangian describing these branes on their world-volume has been constructed. In the previous papers [18–24, 24–31] we refer these models as "F-theory", but we have renamed our formulation from "F-theory" to $\mathcal{A}$-theory in our most recent work [1], in order to present it as a general framework that includes all spacetime dimensions and accommodates all duality symmetries.

Exceptional Field Theory (EFT) applies DFT concepts to exceptional groups [32–44]. The symmetry of exceptional groups was initially discovered as a partial dimensional symmetry of background fields in 11-dimensional supergravity [45, 46]. The generalized diffeomorphism in EFT is characterized by the "Y-tensor", which reflects the structure of the exceptional group. This Y-tensor, $Y^{MN}{}_{PQ}$, is related to the group invariant metric in $\mathcal{A}$-theory, denoted by $\eta^{MNm}$, through the relation $Y^{MN}{}_{PQ} = \eta^{MNm}\eta_{PQm}$ where $M, N, \cdots$ are the spacetime indices and $m$ is the world-volume index. These indices correspond to different representations of the exceptional group. The origin of the Y-tensor lies in the Schwinger term of the current algebra, $\{\rhd_M(\sigma), \rhd_N(\sigma')\} = \eta_{MNm}\partial^m\delta(\sigma - \sigma')$, where the world-volume derivative $\partial^m$ is defined through the commutator with the Virasoro constraint, $\mathcal{S}^m = \frac{1}{2}\eta^{MNm}\rhd_M\rhd_N$. The section condition given by the Y-tensor, $Y^{MN}{}_{PQ}\partial_M\partial_N = 0$, is related to the zero-mode of the Virasoro constraint in $\mathcal{A}$-theory. Specifically, the zero-mode component of the constraint $\mathcal{S}^m$ takes the form $\mathcal{S}^m|_{0\text{-modes}} = \eta^{MNm}\partial_M\partial_N = 0$, establishing a direct connection between the section condition in EFT and the Virasoro structure of $\mathcal{A}$-theory. Active research continues on expressing exceptional groups through brane current algebra [47–54].

This paper focuses on the 5-brane that describes D=3 $\mathcal{A}$-theory with SL(5) U-duality symmetry. In the case of D=3, the theory provides a nontrivial yet tractable example that includes various types of branes and permits explicit computations. The Virasoro algebra, when extended to incorporate brane degrees of freedom, involves Gauß law–type constraints that are intrinsic to the brane. These constraints facilitate the dimensional reduction of the brane world-volume. Such an extended Virasoro algebra serves as a useful prototype for generalization to higher dimensions. Furthermore, the construction of the Lagrangian in this framework constitutes a significant milestone toward extending the formulation to higher-dimensional cases. We clarify how the usual string, and $\mathcal{T}$-string and membrane of 11-dimensional supergravity (M2) emerge. Specifically, we derive the Lagrangians for the conventional non-perturbative M2-brane, the O(D,D)-covariant $\mathcal{T}$-string, and the conventional string from the $\mathcal{A}$-theory 5-brane ($\mathcal{A}$5) Lagrangian.

## 1.2 Summary

In this paper, we focus on the SL(5) U-duality symmetry and clarify the relation between the $\mathcal{A}$-theory five-brane ($\mathcal{A}$5-brane) and conventional branes. The $\mathcal{A}$5-brane is a 5-brane that exhibits manifest SL(5) U-duality symmetry, and it is described by a perturbative Lagrangian described using the SL(5) rank-two antisymmetric tensor coordinate [1, 18, 30]. The $\mathcal{A}$5-brane theory is reduced, through the duality reduction procedure, to the $\mathcal{T}$-string with O(3,3) T-duality symmetry, the $\mathcal{M}$5-brane with GL(4) duality symmetry, and the $S$tring with GL(3) symmetry. The interrelation among these theories forms a diamond-shaped diagram. In this paper, we present the sectioning procedure along this diamond contour. Additionally, we detail reduction procedures from the $\mathcal{T}$-string Lagrangian to the conventional string Lagrangian, and from the $\mathcal{M}$5-brane Lagrangian to the conventional non-perturbative M2-brane Lagrangian [55]. We further generalize the dimensional reduction of S-duality, as applied to 11-dimensional supergravity and its relation to type IIA supergravity [2], to the case of T-duality, as discussed in subsection 1.3. For S-duality, $\lambda_{\text{string}} \leftrightarrow 1/\lambda_{\text{string}}$, the dimensional reduction is realized by taking the limit $\lambda_{\text{string}} \ll 1$ in the metric. For T-duality, characterized by $R/\sqrt{\alpha'} \leftrightarrow \sqrt{\alpha'}/R$ with string length $l_{\text{string}} = \sqrt{\alpha'}$, the dimensional reduction is achieved by taking the large-radius limit $R \gg \sqrt{\alpha'}$, such that the O(D,D) spacetime for the $\mathcal{T}$-string reduces to the conventional D-dimensional spacetime. Finally, we propose a perturbative $\mathcal{M}$5-brane Lagrangian in a supergravity background derived from the $\mathcal{A}$5-brane, as given in equation (109).

In section 2 the relationships among the $\mathcal{A}$5-brane, $\mathcal{M}$5-brane, $\mathcal{T}$-string, and $S$tring theories are elucidated through diamond diagrams based on duality symmetries. The diamond diagram represents a contour in the U-duality plane, which is parametrized by two quantities: the string coupling and a scale defined by the string length. The branes are described in terms of field strengths, where the spacetime coordinates possess the gauge symmetry generated by the Gauß law constraint. These field strengths and their associated gauge parameters, along with the world-volume and spacetime coordinates, transform as representations under the relevant duality symmetries. The world-volume diffeomorphism is generated by the Virasoro constraints $\mathcal{S} = 0$. Dimensions to reduce are determined by solving the Virasoro constraints $\mathcal{S} = 0$ for spacetime and by solving the Gauß law constraints $\mathcal{U} = 0$ for world-volume.

In section 3 both the SL(5) and the SL(6) covariant Lagrangians of the $\mathcal{A}$5-brane are given, where the SL(6) manifests the 5-brane world-volume Lorentz symmetry. Duff and Lu [56,57] showed that the membrane theory exhibits the SL(5) duality symmetry by the Gaillard-Zumino approach. In general, the symmetry of Lagrangian formulation is larger than that of the corresponding Hamiltonian formulation. In $\mathcal{A}$-theory, the U-duality symmetry in the Hamiltonian formulation, G-symmetry, is enhanced to a novel duality symmetry in the Lagrangian formulation, A-symmetry. This symmetry enhancement in higher-dimensional cases (D < 6) is

summarized on page 6 of [22] and page 14 of [1]. It was shown that the brane world-volume metric is also transformed conformally under the SL(5) duality transformation as well as the spacetime background fields in [56,57]. This mixing between spacetime and world-volume is a manifestation of the extended SL(6) duality symmetry transformation. The SL(6) vielbein includes both the SL(5) spacetime vielbein and the 6-dimensional world-volume vielbein, so the spacetime and world-volume are mixed under the new duality symmetry SL(6). The SL(6) formulation is useful to reduce to other branes: Since the string world-sheet directions and the spacetime directions are direct sum, the SL(6) vielbein is in a block diagonal form as shown in subsection 5.1. On the other hand, the brane world-volume directions share the spacetime directions unlike the string as shown in subsection 6.1.

In section 4, we begin with the O(D,D) string Hamiltonian and apply the double zweibein method [58,59] to derive the $\mathcal{T}$-string Lagrangian. We then present the reduction procedure from the O(D,D) $\mathcal{T}$-string Lagrangian to the conventional string Lagrangian in D dimensions, following an approach analogous to that in subsection 1.3. It is shown that the Wess–Zumino term can be obtained by adding a total derivative term.

In section 5, we start from the $\mathcal{A}$5-brane Lagrangian and present the reduction procedure leading to the $\mathcal{T}$-string Lagrangian. The O(D,D) background gauge field is reformulated using SL(4) tensor indices in such a way that it couples naturally to the SL(4) tensor coordinates of the $\mathcal{T}$-string. Subsequently, by applying the procedure described in section 4, we derive the conventional string Lagrangian.

In section 6 we begin with the SL(6) covariant $\mathcal{A}$5-brane Lagrangian which leads to a new perturbative $\mathcal{M}$5-brane Lagrangian. We further reduce it to the conventional M2-brane Lagrangian. The "perturbative" $\mathcal{M}$5-brane Lagrangian is formulated as a bilinear expression in terms of currents, while the "non-perturbative" M2-brane Lagrangian comprises the sum of the Nambu–Goto and Wess–Zumino terms. The dimensional reduction from the $\mathcal{M}$5-brane to the M2-brane is implemented via the "non-perturbative projection" $\partial^m = \epsilon^{ij}\partial_j x^m \partial_i$, in (114) and the gauge fixing of the world-volume metric in (120). The Nambu-Goto Lagrangian is obtained by the gauge choice of the world-volume vielbein, while the Wess-Zumino term is obtained by adding the total derivative term.

## 1.3 Dimensional reduction procedure

In [2], it was pointed out that under an S-duality transformation between the 10-dimensional type IIA theory and the 11-dimensional supergravity theory, the structure of the supersymmetry algebra remains invariant, although the interpretation of the central charge changes. The global superalgebra, involving supercharges $Q$ and $Q'$ of the opposite chirality, 10-dimensional momenta $P$, and a central charge $W$, is given by $\{Q, Q\} \sim P \sim \{Q', Q'\}$, $\{Q_\alpha, Q'_{\dot\beta}\} \sim \delta_{\alpha\dot\beta} W$. The central charge $W$ is interpreted as the Ramond-Ramond (RR) D0-brane charge in 10 dimensions, and as the momentum in the 11th dimension in 11-dimensional supergravity. The 11-dimensional spacetime reduces into the 10-dimensional spacetime in the weak coupling limit $e^{2\phi} \ll 1$,

$$ds_{11}^2 = g_{mn}^{[10]}dx^m dx^n + e^{2\phi}(dy - A_m dx^m)^2 \xrightarrow[\text{reduction}]{\text{dimensional}} ds_{10}^2 = g_{mn}^{[10]}dx^m dx^n, \quad (1)$$

with the 11-th dimensional coordinate $y$ and the string coupling $\lambda_{\text{string}} = e^{3\phi/2}$. The 11-dimensional momentum $W$ is maintained as the D0-brane charge in the 10-dimeniosnal IIA theory after the dimensional reduction.

This framework is generalized to incorporate T-duality. We compare the superalgebra of the 2$D$-dimensional $\mathcal{T}$-string theory, which exhibits manifest T-duality, with the type II superalgebra of conventional string theory in $D$ dimensions. The global type II superalgebra involves two supercharges, $Q$ and $Q'$, and the $D$-dimensional momentum $P$, and is expressed

as $\{Q, Q\} \sim (P + \check{P})$, $\{Q', Q'\} \sim (P - \tilde{P})$, where $\tilde{P}$ is a central charge. In $D$ dimensions, $\tilde{P}$ is interpreted as an NS–NS charge, while in $2D$ dimensions, it corresponds to the additional momenta associated with the extended spacetime. By restoring the $\alpha'$ dependence in the $O(D, D)$ momentum–winding vector, $(p_m, \frac{1}{\alpha'}\partial_\sigma x^m) \to (p_m, \frac{1}{\alpha'}\tilde{p}^m)$, the canonical conjugate coordinates become $(x^m, \alpha' y_m)$. At small compactification scales $(R \ll \sqrt{\alpha'})$, the winding modes become light and are readily excited, whereas at large scales $(R \gg \sqrt{\alpha'})$, they become heavy and only the momentum modes remain dynamically relevant. Manifest T-duality is broken by choosing a specific background such that the $2D$-dimensional spacetime effectively reduces to the $D$-dimensional one in the limit $R \gg \sqrt{\alpha'}$:

$$ds_{2D}^2 = g_{mn}dx^m dx^n + \alpha'^2(dy_m - dx^l B_{lm})g^{mn}(dy_n - dx^k B_{kn}) \xrightarrow[\text{reduction}]{\text{dimensional}} ds_D^2 = g_{mn}dx^m dx^n.$$
(2)

Here, $y_m$ denotes the additional $D$-dimensional coordinates, and $B_{mn}$ is the NS–NS gauge field. The additional momenta in the extended dimensions are preserved as NS–NS charges after the dimensional reduction.

This dimensional reduction procedure corresponds to the gauge fixing of the dimensional reduction constraint which is the first class constraint in Hamiltonian formulation. The dimensional reduction constraint and the gauge fixing to reduce the conventional string are discussed in [60] for a flat space case. The dimensional reduction constraint is the $y$ component of the symmetry generator, $\tilde{\triangleright}_y = 0$. The gauge fixing condition $\partial_\sigma y = 0$ reduces the set of conventional string operators, the physical momentum $P_x \neq 0$ and left/right covariant derivatives $P_x \pm \partial_\sigma x$. In Lagrangian formulation the momentum is replaced by $P_X = \partial L / \partial \dot{X}$. It is generalized to the brane case, and then the dimensional reduction constraint turns out to be the Virasoro constraint in which one of the momenta is replaced by the 0-mode [19]. For zero-mode momenta $p_x$, $p_y$ and momenta including all modes $P_x$, $P_y$, the dimensional reduction constraint $\tilde{\triangleright}_y = 0$ is expressed by the Virasoro operator as $p_y \cdot P_x + p_x \cdot P_y = 0 \to P_y = 0$.

Although the equation of motion derived from the doubled Lagrangian, when combined with the self-duality condition, coincides with that obtained from the original Lagrangian, the self-duality condition causes the doubled Lagrangian to vanish [61]. In particular, the self-duality condition $\partial_\mu x = \epsilon_{\mu\nu}\partial^\nu y$ reduces the Lagrangian of the $O(D, D)$ $\mathcal{T}$-string in flat space to zero, as follows

$$L = \frac{1}{2}\left(\dot{x}^2 - x'^2 + \dot{y}^2 - y'^2\right) \xrightarrow{\text{selfduality}} 0.$$

It is also mentioned that the naive section $y = 0$ the $\mathcal{T}$-string Lagrangian in curved background does not reduce to the expected string Lagrangian in curved background as

$$L = \frac{1}{2}(\partial_+ x^m \ \partial_+ y_m)\begin{pmatrix} g_{mn} - B_{ml}g^{lk}B_{kn} & -B_{ml}g^{ln} \\ g^{ml}B_{ln} & g^{mn} \end{pmatrix}\begin{pmatrix} \partial_- x^n \\ \partial_- y_n \end{pmatrix}$$

$$\xrightarrow{y=0} \frac{1}{2}\partial_+ x^m \left(g_{mn} - B_{ml}g^{lk}B_{kn}\right)\partial_- x^n.$$

The following points are also noteworthy. Integrating out the $(dy + \cdots)^2$ term is possible in the case of a constant background. However, for a general non-constant background, the path integral over $dy$ in $\exp\left[-\int (dy + \cdots)g(dy + \cdots)\right]$ yields a Jacobian factor $\sqrt{g}$ in the path integral measure, which in turn generates an additional term in the effective action. Using the equation of motion is again valid in the constant background case, but it does not reproduce the conventional string Lagrangian when the background fields $g(x, y)$ and $B(x, y)$ are non-constant. Furthermore, imposing both conditions $\partial_+ y - \partial_+ x B = 0$ and $\partial_- y - \partial_- x B = 0$ is inconsistent, since the integrability condition is violated in curved backgrounds where $[\partial_+, \partial_-]y \neq 0$. Several studies have been devoted to refining the reduction to the conventional string Lagrangian, resulting in a variety of interesting approaches [43, 50, 61–64].

Instead we propose the reduction procedure from $\mathcal{T}$-string Lagrangian to the conventional string Lagrangian: (1) adding the total derivative term $-\partial_\mu(\epsilon^{\mu\nu}x^m\partial_\nu y_m)$ to derive the Wess-Zumino term, then (2) the dimensional reduction (2) as

$$
L = \frac{1}{2}(\partial_+ x^m \ \ \partial_+ y_m)\begin{pmatrix} g_{mn}-B_{ml}g^{lk}B_{kn} & -B_{ml}g^{ln} \\ g^{ml}B_{ln} & g^{mn} \end{pmatrix}\begin{pmatrix} \partial_- x^n \\ \partial_- y_n \end{pmatrix}-\partial_\mu(\epsilon^{\mu\nu}x^m\partial_\nu y_m)
$$
$$
\xrightarrow[\text{reduction (1.2)}]{\text{dimensional}} \partial_+ x^m(g_{mn}+B_{mn})\partial_- x^n\,. \tag{3}
$$

This is the expected string Lagrangian up to the normalization factor two which can be absorbed by the Lagrange multiplier. The section conditions of spacetime fields $\Phi(x,y)$ are consistent with the Lagrangian where the section $y=0$ can be chosen as $\Phi(x)$.

This procedure is similar to the usual dimensional reduction where the reduction is done in the local flat Lorentz coordinate. i.e. Suppose that we have a line element $dx^A \equiv dx^M E_M{}^A$. We decompose the doubled coordinate $dx^A$ into $dx^a$ and $dy_a$ in the local Lorenz frame, and then discard $(dy_a)^2$. Since the metric ($\hat{\eta}$-tensor) in local flat spacetime is already diagonal, in practice we can just apply this reduction by deleting certain blocks of $\hat{\eta}$-tensor similar to (2).

The main purpose of this paper is to carry out the above reduction procedure in several specific cases. In general, however, the procedure can be schematically summarized as follows.

1. We start with the current algebra defined on an extended space of coordinates, where both momentum and winding modes have their corresponding conjugate coordinates. The Hamiltonian is written as a sum of self-dual and anti-self-dual constraints: $H = g\mathcal{H} + \tilde{g}\tilde{\mathcal{H}} + s_m\mathcal{S}^m + \tilde{s}_m\tilde{\mathcal{S}}^m + Y^m\mathcal{U}_m$, where $\mathcal{H},\tilde{\mathcal{H}}$ are the $\tau$ component of the Virasoro constraints and its dual counterpart, $\mathcal{S}^m,\tilde{\mathcal{S}}^m$ are the $\sigma^m$ components of the Virasoro constraints and its dual counterpart. $\mathcal{U}_m$ is the Gauß law constraint specific to branes.

2. The Lagrangian is obtained via a Legendre transformation of the Hamiltonian $H$. This has been performed in previous works for various theories [1]. Schematically, the Lagrangian takes the form $L = \Phi J_{\text{SD}}\cdot\hat{\eta}\cdot J_{\overline{\text{SD}}} + \Lambda\cdot J_{\overline{\text{SD}}}\cdot\hat{\eta}\cdot J_{\overline{\text{SD}}} + \cdots$, where $J^A_{\text{SD}/\overline{\text{SD}}}$ are the selfdual and anti-selfdual currents. They are coupled with vielbein, and thus they have flat indices. $\Phi$ and $\Lambda$ are Lagrange multipliers which are functions of $g,s^m,\tilde{g},\tilde{s}^m$. One can gauge fix $\Lambda = 0$ by the suitable choice of original parameters.

3. Separate coordinates and currents into the physical part and the auxiliary part as $X^M \to x^m, y^\mu$, and $J^A \to J^a, J^\alpha$ where $x^m$ represents the physical coordinates for the target string or brane theory, and $y^\mu$ denotes auxiliary coordinates. Dimensional reduction is then performed according to equation (2).

4. The reduced Lagrangian $L' = J^a{}_{\text{SD}}\hat{\eta}_{ab}J^b{}_{\overline{\text{SD}}}$ can be shown to reproduce the desired string or brane action, up to the absence of the Wess–Zumino (WZ) term. We find that adding a total derivative term to the Lagrangian restores the WZ term

$$
L + \text{Total derivative} = J^a{}_{\text{SD}}\hat{\eta}_{ab}J^b{}_{\overline{\text{SD}}} + \tilde{J}^\alpha_{\text{SD}}\hat{\eta}_{\alpha\beta}\tilde{J}^\beta{}_{\overline{\text{SD}}} + L_{\text{WZ}}\,.
$$

Here, the current $\tilde{J}^\alpha$ is modified by the addition of the total derivative term and is subsequently eliminated through the dimensional reduction (2). This procedure yields the correct WZ term.

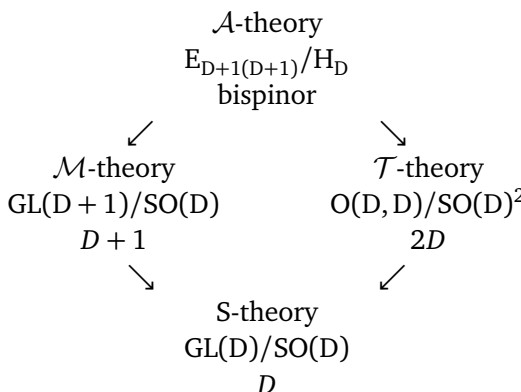

Figure 1: $G$-symmetries of D $= D$ theories and spacetime dimensions.

## 2 Theories with manifest duality symmetries and sectionings

### 2.1 Diamond diagrams

The duality web of the $G$-symmetry in $\mathcal{A}$-theory is represented by the diamond diagram shown in Fig. 1, as studied in [19]. The $G$-symmetry, associated with the coset group $G/H$, plays the role of a duality symmetry. The coset parameter serves as the gauge field of the duality-covariant geometry, incorporating the spacetime vielbein as well as the NS–NS and R–R gauge fields of superstring theory.

The relationships among these duality groups are illustrated using Dynkin diagrams, as discussed in [1]. Removing a single node from the Dynkin diagram of $E_{D+1(D+1)}$ reduces it to that of either GL(D+1) or O(D,D), depending on which node is removed. Further removing one more node from the Dynkin diagram of GL(D+1) or O(D,D) leads to that of GL(D).

In this paper we focus on D=3 case where the $G$-symmetry is SL(5) and the diamond diagram becomes Fig. 2. This SL(5) duality symmetry is enlarged to SL(6) for the (5+1)-dimensional world-volume covariance in Lagrangian [1]. We named this enlarged symmetry "$A$-symmetry". This $\mathcal{A}$-theory unifies the spacetime and the world-volume, in a sense that the coset parameter of $A/L$=SL(6)/GL(4) includes not only the spacetime vielbein field but also the world-volume vielbein field.

In this paper, we focus on the D=3 case, where the $G$-symmetry is SL(5) and the diamond diagram corresponds to Fig. 2. This SL(5) duality symmetry is further enhanced to SL(6) in order to accommodate the (5+1)-dimensional world-volume covariance in the Lagrangian formulation [1]. We refer to this enlarged symmetry as the "$A$-symmetry". The resulting $\mathcal{A}$-theory unifies the spacetime and world-volume structures, in the sense that the coset parameter of $A/L$ =SL(6)/GL(4) includes not only the spacetime vielbein field but also the world-volume vielbein field. We note that $H = $ SO(D) is used instead of SO(D-1,1) for simplicity. Consequently, a Wick rotation is required to properly account for the time component in this section and elsewhere.

### 2.2 Representations

In duality covariant theories, spacetime and world-volume coordinates transform as representations of the duality symmetry ($A$-symmetry or $G$-symmetry), which determines the world-volume dimension. The Gauß law constraint generates gauge symmetry of the duality covariant spacetime coordinate, making the brane current correspond to a field strength.

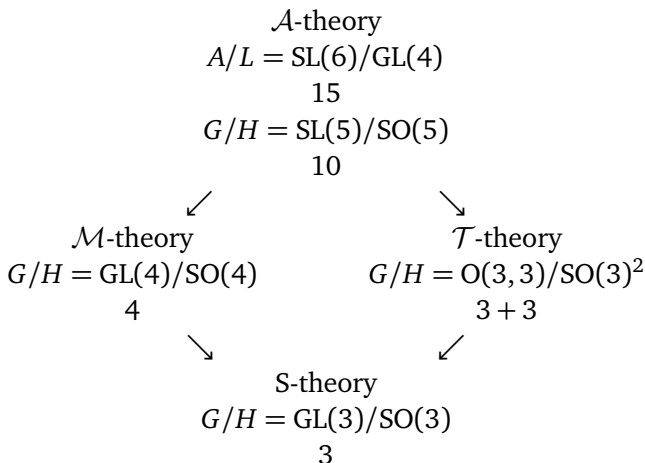

Figure 2: *A*- and *G*-symmetries of D = 3 theories and spacetime dimensions.

The D=3 $\mathcal{M}$, $\mathcal{T}$, $S$-theories are obtained from the D=3 $\mathcal{A}$-theory [18, 30]. We list representations of duality groups in Tab. 1; the world-volume derivative $\partial^m$, the gauge parameter $\lambda^m$, the spacetime coordinate $X^M$, and the field strength (the current) $F_M = \eta_{MNm} \partial^m X^N$ ($J_\mu{}^M = \partial_\mu X^M$, $\mu = (\tau, \sigma)$). $\eta_{MNm}$ is the *G*-symmetry invariant tensor which enters the current algebra, where the SL(5) invariant metric is $\eta_{MNl} = \eta_{m_1 m_2 n_1 n_2 l} = \epsilon_{m_1 m_2 n_1 n_2 l}$. The world-volume dimension of $\mathcal{M}$-theory is still $1 \oplus 5$ where four dimensions are embedded in the 4 spacetime $x^{\underline{m}}$ and one dimension is embedded in the internal space, so we denote as $1 \oplus 4(5)$.

The field strengths and currents together with the gauge transformations are given concretely as follows.

1. $\mathcal{A}$5-brane field strengths

   (a) World-volume covariant $\mathcal{A}$5-brane field strength

   The SL(6) *A*-symmetry covariant $\mathcal{A}$-theory is described by a 5-brane with the manifest SL(6) new duality symmetry which manifests 6-dimensional world-volume Lorentz symmetry, namely world-volume covariant $\mathcal{A}$5-brane.

   $$F^{\hat{m}\hat{n}\hat{p}} = \frac{1}{2} \partial^{[\hat{m}} X^{\hat{n}\hat{p}]}, \qquad \delta_\lambda X^{\hat{m}\hat{n}} = \partial^{[\hat{m}} \lambda^{\hat{n}]}, \qquad \hat{m} = 0, 1, \ldots, 5. \tag{4}$$

   (b) $\mathcal{A}$5-brane field strength

   The SL(5) *G*-symmetry covariant $\mathcal{A}$-theory is described by a 5-brane with manifest SL(5) U-duality symmetry, namely $\mathcal{A}$5-brane.

   $$\begin{cases} F_\tau{}^{mn} = \dot{X}^{mn} - \partial^{[m} Y^{n]}, \\ F_{\sigma; m_1 m_2} = \frac{1}{2} \epsilon_{m_1 \cdots m_5} \partial^{m_3} X^{m_4 m_5}, \\ m = 1, \ldots, 5, \end{cases} \qquad \begin{cases} \delta_\lambda X^{mn} = \partial^{[m} \lambda^{n]}, \\ \delta_\lambda Y^m = \dot{\lambda}^m - \partial^m \lambda^0. \end{cases} \tag{5}$$

2. $\mathcal{M}$5-brane field strength

   The GL(4) $\mathcal{M}$-theory is described by a 5-brane with the manifest GL(4) duality symmetry, namely $\mathcal{M}$5-brane. We focus only on 4-dimensional subspace of the 5-dimensional

Table 1: Representations of the duality groups of theories.

| Theories Groups | World-volumes $\partial$ | Gauges $\lambda$ | Spacetimes $X$ | Field strengths $F$ |
|---|---|---|---|---|
| $\mathcal{A}$-theory SL(6) | 6 $\partial^{\hat{m}},\ _{\hat{m}=0,\cdots,5}$ | 6 $\lambda^{\hat{m}}$ | 15 $X^{\hat{m}\hat{n}}$ | 20 $F^{\hat{m}\hat{n}\hat{p}}$ |
| $\mathcal{A}$-theory SL(5) | $1\oplus 5$ $\partial^{0},\partial^{m},\ _{m=1,\cdots,5}$ | $1\oplus 5$ $\lambda^{0},\lambda^{m}$ | $10\oplus 5$ $X^{mn},Y^{m}$ | $10\oplus 10'$ $F_{\tau}{}^{mn},F_{\sigma mn}$ |
| $\mathcal{M}$-theory GL(4) | $1\oplus 4(5)$ $\partial^{0},\partial^{\underline{m}},\ _{\underline{m}=1,\cdots,4}$ | $1\oplus 4$ $\lambda^{0},\lambda^{\underline{m}}$ | $4\oplus 1$ $x^{\underline{m}},Y$ | $4\oplus 6$ $F_{\tau}{}^{\underline{m}},F_{\sigma\underline{mn}}$ |
| $\mathcal{T}$-theory O(3,3) | $1\oplus 1$ $\partial^{0},\partial^{\sigma},\ _{\bar{m}=1,2,3}$ | 0 | $3\oplus 3'$ $x^{\bar{m}},y^{\bar{m}\bar{n}}$ | $3\oplus 3\oplus 3'\oplus 3'$ $J_{\tau}{}^{\bar{m}},J_{\sigma\bar{m}\bar{n}};J_{\tau}{}^{\bar{m}},J_{\sigma\bar{m}},$ |
| $S$-theory GL(3) | $1\oplus 1$ $\partial^{0},\partial^{\sigma},\ _{\bar{m}=1,2,3}$ | 0 | 3 $x^{\bar{m}}$ | $3\oplus 3$ $J_{\tau}{}^{\bar{m}},J_{\sigma\bar{m}\bar{n}}$ |

world-volume which is embedded in the 4-dimensional spacetime. This $\mathcal{M}5$-brane extends over both the main space (i.e., the duality-covariant space) and the internal space. Four of its world-volume directions lie in the main space while the remaining directions lie in the internal space, specifically one world-volume direction in the Hamiltonian formalism, or two in the Lagrangian formalism. Considering the critical string action in the full spacetime structure is an interesting subject, although it lies beyond the scope of the present discussion. The relationship between the main space and the internal space is schematically illustrated in Figure 2 the "slug diagram" (see page 27 in [20] or page 14 in [1]). In the case of D = 3 the main space coordinate is represented by a bispinor $X^{\alpha\beta}$, and the world-volume coordinate by an antisymmetric bispinor $\sigma^{[\alpha\beta]}$, with $\alpha = 1,\ldots,4$. The internal space coordinate is given by a bispinor $Y^{[\alpha'\beta']}$, where $\alpha' = 1,\ldots,8$. The total number of supersymmetries is 32, which corresponds to the product of the dimensions of the spinor indices $32 = 4\times 8$. It is noted that the assignment of the duality symmetric space in $\mathcal{A}$-theory differs from that in conventional formulations. In $\mathcal{A}$-theory, the duality-symmetric space is assigned to the main "spacetime" rather than the internal space, such that all tensor gauge fields are automatically incorporated into the coset parameter of $E_{D+1}/H$.

Physical currents are as follows.

$$\begin{cases} F_{\tau}{}^{\underline{m}} = \dot{x}^{\underline{m}} + \partial^{\underline{m}}Y\,, \\ F_{\sigma;\underline{m_1 m_2}} = -\epsilon_{\underline{m_1}\cdots\underline{m_4}}\partial^{\underline{m_3}}x^{\underline{m_4}}\,, \\ \underline{m} = 1,\ldots,4\,, \end{cases} \qquad \begin{cases} \delta_\lambda x^{\underline{m}} = \partial^{\underline{m}}\lambda\,, \\ \delta_\lambda Y = -\dot{\lambda}\,. \end{cases} \tag{6}$$

The following currents are auxiliary written by auxiliary coordinates $y^{\underline{mn}}$, $Y^{\underline{m}}$.

$$\begin{cases} F_\tau{}^{\underline{mn}} = \dot{y}^{\underline{mn}} - \partial^{[\underline{m}}Y^{\underline{n}]}, \\ F_{\sigma;\underline{m}_1} = \frac{1}{2}\epsilon_{\underline{m}_1\cdots\underline{m}_4}\partial^{\underline{m}_2}y^{\underline{m}_3\underline{m}_4}, \end{cases} \qquad \begin{cases} \delta_\lambda y^{\underline{mn}} = \partial^{[\underline{m}}\lambda^{\underline{n}]}, \\ \delta_\lambda Y^{\underline{m}} = \dot{\lambda}^{\underline{m}}, \\ \delta_\lambda \lambda^{\underline{m}} = \partial^{\underline{m}}\lambda. \end{cases} \tag{7}$$

These currents constitute the SL(5) $A$-symmetry together with (6), and they are used to lead the non-perturbative M2-brane Lagrangian.

3. $\mathcal{T}$-string currents

The O(3,3) $\mathcal{T}$-theory is described by a string with the manifest O(3,3) T-duality symmetry, namely $\mathcal{T}$-string.

$$\begin{cases} J_\tau{}^{\underline{m}_1\underline{m}_2} = \dot{X}^{\underline{m}_1\underline{m}_2}, \\ J_{\sigma\underline{m}_1\underline{m}_2} = \frac{1}{2}\epsilon_{\underline{m}_1\cdots\underline{m}_4}\partial_\sigma X^{\underline{m}_3\underline{m}_4}. \end{cases} \tag{8}$$

It is convenient to represent in terms of $x^{\bar{m}}$ and $y_{\bar{m}} = \frac{1}{2}\epsilon_{\bar{m}\bar{n}\bar{l}}y^{\bar{n}\bar{l}}$.

$$\begin{cases} J_\tau{}^{\bar{m}} = \dot{x}^{\bar{m}}, \\ J_{\sigma\bar{m}_1\bar{m}_2} = -\epsilon_{\bar{m}_1\bar{m}_2\bar{m}_3}\partial_\sigma x^{\bar{m}_3}, \\ J_\tau{}^{\bar{m}_1\bar{m}_2} = \dot{y}^{\bar{m}_1\bar{m}_2}, \\ J_{\sigma\bar{m}_1} = \frac{1}{2}\epsilon_{\bar{m}_1\bar{m}_2\bar{m}_3}\partial_\sigma y^{\bar{m}_2\bar{m}_3}, \\ \bar{m} = 1,2,3. \end{cases} \tag{9}$$

4. $S$-string currents

The GL(3) $S$-theory is described by a string with the manifest GL(3) spacetime diffeomorphism symmetry, namely a 3-dimensional string.

$$\begin{cases} J_\tau{}^{\bar{m}} = \dot{x}^{\bar{m}}, \\ J_{\sigma\bar{m}_1\bar{m}_2} = -\epsilon_{\bar{m}_1\bar{m}_2\bar{m}_3}\partial_\sigma x^{\bar{m}_3}. \end{cases} \tag{10}$$

Some minus signs come from the mere notation $\epsilon_{1234} = 1 = -\epsilon_{4123}$. It is denoted that these currents are flat currents, and in later sections flat current symbols $\mathring{F}$ or $\mathring{J}$ will be used to distinguish from curved background currents.

## 2.3 Constraints and sectionings

The theories in Hamiltonian formulation are constructed by the current algebra with manifest duality symmetries [18]. The Spacetime translation is generated by the covariant derivative $\triangleright_M(\sigma)$. The $p$-brane current algebra with $G$-symmetry covariance is given by

$$\left[\triangleright_M(\sigma), \triangleright_N(\sigma')\right] = 2if_{MN}{}^L \triangleright_L(\sigma)\delta(\sigma-\sigma') + 2i\eta_{MNm}\partial^m\delta^{(p)}(\sigma-\sigma'). \tag{11}$$

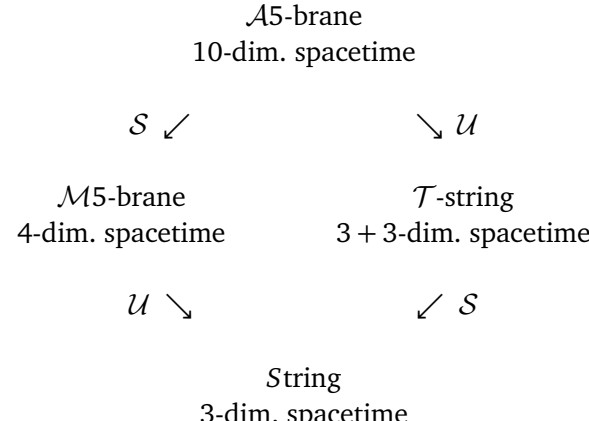

$\mathcal{A}$5-brane
10-dim. spacetime

$\mathcal{S}$ ↙      ↘ $\mathcal{U}$

$\mathcal{M}$5-brane          $\mathcal{T}$-string
4-dim. spacetime          3 + 3-dim. spacetime

$\mathcal{U}$ ↘      ↙ $\mathcal{S}$

$\mathcal{S}$tring
3-dim. spacetime

Figure 3: Diamond diagram of Sectionings of branes of D = 3 theories.

Branes are governed by the brane Virasoro constraints $\mathcal{S}^m = \frac{1}{2} \triangleright_M \eta^{MNm} \triangleright_N = 0$ and $\mathcal{H} = \frac{1}{2} \triangleright_M \hat{\eta}^{MN} \triangleright_N = 0$ together with the Gauß law constraints $\mathcal{U}_m = 0$ which is required by the closure of the Virasoro algebra. $\hat{\eta}^{MN}$ is the $H$-invariant metric. Theories are related by sectionings; The Virasoro constraint $\mathcal{S}^m = 0$ gives the section conditions to reduce the spacetime dimensions, and the Gauß law constraint $\mathcal{U}_m = 0$ is used to reduce the world-volume dimension as Fig. 3.

The spacetime covariant derivatives, constraints and section conditions are given [18,30] in Fig. 3 concretely as follows.

1. $\mathcal{A}$5-brane in 10-dimensional spacetime

    The 10-dimensional spacetime is described by the rank-two anti-symmetric tensor covariant derivative $\triangleright_{m_1 m_2}$ as

$$\triangleright_{m_1 m_2} = P_{m_1 m_2} + \frac{1}{2} \epsilon_{m_1 \cdots m_5} \partial^{m_3} X^{m_4 m_5} \,, \tag{12}$$

    where $P_{mn}$ is canonical conjugate of $X^{mn}$ with $[P_{mn}(\sigma), X^{lk}(\sigma')] = \frac{1}{i} \delta_m^{[l} \delta_n^{k]} \delta^{(5)}(\sigma - \sigma')$ and $m = 1, \ldots, 5$.

    The SL(5) covariant current algebra of $\mathcal{A}$5-brane is given by

$$\left[ \triangleright_{m_1 m_2}(\sigma), \triangleright_{m_3 m_4}(\sigma') \right] = 2i \epsilon_{m_1 \cdots m_5} \partial^{m_5} \delta^{(5)}(\sigma - \sigma') \,. \tag{13}$$

    The 5-dimensional world-volume diffeomorphism is generated by the Virasoro constraints $\mathcal{S}^m = 0$ while the world-volume time diffeomorphism is generated by $\mathcal{H} = 0$. The Gauß law constraint $\mathcal{U}_m = 0$ generates the gauge symmetry of the spacetime coordinate. These constraints are given by [1] as:

$$\begin{cases} \mathcal{S}^m = \frac{1}{16} \triangleright_{m_1 m_2} \epsilon^{mm_1 \cdots m_4} \triangleright_{m_3 m_4} = 0 \,, \\ \mathcal{H} = \frac{1}{16} \triangleright_{m_1 m_2} \delta^{m_1 [n_1} \delta^{n_2] m_2} \triangleright_{n_1 n_2} = 0 \,, \\ \mathcal{U}_m = \partial^n \triangleright_{mn} = 0 \,. \end{cases} \tag{14}$$

    The SL(5) covariant constraints $\mathcal{S}^m = 0$ and $\mathcal{U}_m = 0$ are background independent. These constraints are used as the dimensional reduction and section condition by replacing

Table 2: Constraints, dimensional reduction conditions and section conditions of $\mathcal{A}5$-brane.

|  | Virasoro : $\mathcal{S}^m$ | Gauß law : $\mathcal{U}_m$ |
|---|---|---|
| dimensional reduction | $\epsilon^{mm_1\cdots m_4} P_{m_1 m_2}(\sigma) \dfrac{\partial}{\partial X_0^{m_3 m_4}}$ | $P_{mn}(\sigma) \dfrac{\partial}{\partial \sigma_n}$ |
| section condition | $\epsilon^{mm_1\cdots m_4} \dfrac{\partial}{\partial X_0^{m_1 m_2}} \dfrac{\partial}{\partial X_0^{m_3 m_4}}$ | $\dfrac{\partial}{\partial \sigma_n} \dfrac{\partial}{\partial X_0^{mn}}$ |

the spacetime momentum $P_M(\sigma)$ with the derivative of the 0-mode of the spacetime coordinate $X_0^M$. These operators act on fields $\Phi(X)$ and $\Psi(X)$ as

$$
\begin{aligned}
\frac{\partial}{\partial X_0^M} \frac{\partial}{\partial X_0^N} \Phi(X_0) = 0 &= \frac{\partial}{\partial X_0^M} \Phi(X_0) \frac{\partial}{\partial X_0^N} \Psi(X_0), \\
\frac{\partial}{\partial \sigma_n} \frac{\partial}{\partial X^M} \Phi(\sigma, X(\sigma)) = 0 &= \frac{\partial}{\partial \sigma_n} \Phi(\sigma, X(\sigma)) \frac{\partial}{\partial X^M} \Psi(\sigma, X(\sigma)),
\end{aligned}
\tag{15}
$$

where fields may be functions on $\sigma$ as $\Phi(\sigma, X(\sigma))$ and $\Psi(\sigma, X(\sigma))$.

2. $\mathcal{M}5$-brane in 4-dimensional spacetime

The dimensional reduction of the spacetime is obtained by solving the Virasoro constraint in Tab. 2 as

$$
P_{\underline{mn}}(\sigma) = 0, \quad \underline{m} = 1, \ldots, 4 \qquad \Rightarrow \qquad \epsilon^{mm_1\cdots m_4} \frac{\partial}{\partial X_0^{m_1 m_2}} P_{m_3 m_4}(\sigma) = 0.
\tag{16}
$$

This condition makes $X^{\underline{m_1 m_2}} = y^{\underline{m_1 m_2}}$ to be non-dynamical and reduced dimensionally. The remaining spacetime is 4 dimensions $P_{5\underline{m}} = p_{\underline{m}} \neq 0$.

The 4-dimensional spacetime is described by the covariant derivative $\triangleright_{\underline{m}}$. The 6-dimensional covariant derivative $\triangleright_{\underline{mn}}$ is maintained to construct SL(5) current algebra

$$
\begin{cases}
\triangleright_{\underline{m}} = p_{\underline{m}}, \\[2mm]
\triangleright_{\underline{m_1 m_2}} = -\epsilon_{\underline{m_1}\cdots \underline{m_4}} \partial^{\underline{m_3}} x^{\underline{m_4}},
\end{cases}
\tag{17}
$$

with $X^{5\underline{m}} = x^{\underline{m}}$ and $P_{5\underline{m}} = p_{\underline{m}}$ which is not confused with the 0-mode momentum.

The SL(5) current algebra of $\mathcal{M}5$-brane is

$$
\begin{cases}
\left[ \triangleright_{\underline{m}}(\sigma), \triangleright_{\underline{n}}(\sigma') \right] = 0, \\[2mm]
\left[ \triangleright_{\underline{m_1}}(\sigma), \triangleright_{\underline{m_2 m_3}}(\sigma') \right] = 2i\epsilon_{\underline{m_1}\cdots \underline{m_4}} \partial^{\underline{m_4}} \delta^{(5)}(\sigma - \sigma'), \\[2mm]
\left[ \triangleright_{\underline{m_1 m_2}}(\sigma), \triangleright_{\underline{m_3 m_4}}(\sigma') \right] = 0,
\end{cases}
\tag{18}
$$

where the last algebra forces to $\partial^5 = 0$.

Table 3: Constraints, dimensional reduction conditions and section conditions of $\mathcal{M}5$-brane.

|  | Virasoro : $\mathcal{S}^{\underline{m}}$ | Gauß law : $\mathcal{U}_m$ |
|---|---|---|
| dimensional reduction | $\frac{1}{2}p_{\underline{m}}(\sigma)(\partial^{[\underline{m}}x^{\underline{n}]})$ | $p_{\underline{m}}(\sigma)\partial^{\underline{m}}$ |
| section condition | none | $\partial^{\underline{m}}\dfrac{\partial}{\partial x_0^{\underline{m}}}$ |

The Virasoro operators of $\mathcal{M}5$-brane are

$$
\begin{cases}
\mathcal{S}^{\underline{m}} = \frac{1}{2}\partial^{[\underline{m}}x^{\underline{n}]}p_{\underline{n}}\,, \\[4pt]
\mathcal{S}^5 = \frac{1}{4}\epsilon_{\underline{m}_1\cdots\underline{m}_4}(\partial^{\underline{m}_1}x^{\underline{m}_2})(\partial^{\underline{m}_3}x^{\underline{m}_4})\,, \\[4pt]
\mathcal{H} = \frac{1}{4}p_{\underline{m}}\hat{\eta}^{\underline{mn}}p_{\underline{n}} + \frac{1}{16}(\partial^{[\underline{m}_1}x^{\underline{m}_2]})\hat{\eta}_{\underline{m}_1[\underline{n}_1}\hat{\eta}_{\underline{n}_2]\underline{m}_2}(\partial^{[\underline{n}_1}x^{\underline{n}_2]})\,, \\[4pt]
\mathcal{U}_5 = \partial^{\underline{n}}p_{\underline{n}}\,.
\end{cases}
\tag{19}
$$

These constraints lead to the following dimensional reductions and section conditions.

3. $\mathcal{T}$-string in 6-dimensional spacetime

The dimensional reduction condition of the world-volume is obtained by solving the Gauß law constraint in Tab. 2 as

$$
\partial^{\underline{n}} = 0\,, \quad P_{\underline{m}}(\sigma) = 0 \qquad \Rightarrow \qquad \partial^n P_{mn}(\sigma) = 0\,.
\tag{20}
$$

These conditions make $\partial^5 = \frac{\partial}{\partial\sigma} \neq 0$ and $X^{\underline{m}}$ to be non-dynamical (constant). The remaining spacetime is 6 dimensional $P_{\underline{mn}} = p_{\underline{mn}} \neq 0$.

The 6-dimensional spacetime is described by the covariant derivative $\triangleright_{\underline{mn}}$. The 4-dimensional covariant derivative vanishes $\triangleright_{\underline{m}} = 0$

$$
\triangleright_{\underline{m}_1\underline{m}_2} = p_{\underline{m}_1\underline{m}_2} + \frac{1}{2}\epsilon_{\underline{m}_1\cdots\underline{m}_4}\partial^5 x^{\underline{m}_3\underline{m}_4}\,,
\tag{21}
$$

with $X^{\underline{mn}} = x^{\underline{mn}}$.

The O(3,3) current algebra of $\mathcal{T}$-string is

$$
\left[\triangleright_{\underline{m}_1\underline{m}_2}(\sigma), \triangleright_{\underline{m}_3\underline{m}_4}(\sigma')\right] = 2i\epsilon_{\underline{m}_1\cdots\underline{m}_4}\partial_\sigma\delta(\sigma-\sigma')\,,
$$

with $\partial^5 = \partial_\sigma$.

The Virasoro operators of $\mathcal{T}$-string are

$$
\begin{cases}
\mathcal{S}^5 = \frac{1}{16}\triangleright_{\underline{m}_1\underline{m}_2}\epsilon^{\underline{m}_1\cdots\underline{m}_4}\triangleright_{\underline{m}_3\underline{m}_4}\,, \\[4pt]
\mathcal{H} = \frac{1}{16}\triangleright_{\underline{m}_1\underline{m}_2}\hat{\eta}^{\underline{m}_1[\underline{n}_1}\hat{\eta}^{\underline{n}_2]\underline{m}_2}\triangleright_{\underline{n}_1\underline{n}_2}\,,
\end{cases}
\tag{22}
$$

with $\mathcal{S}^{\underline{m}} = 0 = \mathcal{U}_m$.

The Virasoro constraint $\mathcal{S}^5 = 0$ leads to the following dimensional reduction and the section condition.

4. *String* in 3-dimensional spacetime

   (a) From $\mathcal{T}$-string to *String*

   The dimensional reduction of the spacetime is obtained by solving the Virasoro constraint in Tab. 4 as

   $$P_{\bar{m}\bar{n}}(\sigma) = 0, \quad \bar{m} = 1, 2, 3 \qquad \Rightarrow \qquad P_{\underline{m_1 m_2}}(\sigma) \epsilon^{\underline{m_1} \cdots \underline{m_4}} \frac{\partial}{\partial x_0^{\underline{m_3 m_4}}} = 0. \tag{23}$$

   This condition makes $X^{\bar{m}_1 \bar{m}_2} = y^{\bar{m}_1 \bar{m}_2}$ to be non-dynamical (constant). The remaining spacetime is 3 dimensions $P_{4\bar{m}} = p_{\bar{m}} \neq 0$.

   (b) From $\mathcal{M}5$-brane to *String*

   The dimensional reduction condition of the world-volume is obtained by solving the Gauß law constraint in Tab. 3 as

   $$\partial^{\bar{n}} = 0, \quad P_4(\sigma) = 0 \qquad \Rightarrow \qquad P_{\underline{m}} \partial^{\underline{m}} = 0. \tag{24}$$

   In the 4-dimensional spacetime $\partial^5$ is considered to be 0. These conditions make $\partial^4 = \frac{\partial}{\partial \sigma} = \partial_\sigma \neq 0$ and $X^{54}$ to be non-dynamical (constant). The remaining spacetime is 3 dimensions $P_{4\bar{m}} = p_{\bar{m}} \neq 0$.

The 3-dimensional spacetime is described by the covariant derivative $\triangleright_{4\bar{m}}$. The 3-dimensional covariant derivative vanishes $\triangleright_{\bar{m}_1 \bar{m}_2}$

$$\begin{cases} \triangleright_{\bar{m}} = p_{\bar{m}}, \\ \triangleright_{\bar{m}_1 \bar{m}_2} = \epsilon_{\bar{m}_1 \bar{m}_2 \bar{m}} \partial_\sigma x^{\bar{m}}, \end{cases} \tag{25}$$

with $X^{4\bar{m}} = x^{\bar{m}}$ and $\sigma^5 = \sigma$ via $\mathcal{T}$-string and $X^{5\bar{m}} = x^{\bar{m}}$ and $\sigma^4 = \sigma$ via $\mathcal{M}5$-brane.

The GL(3) current algebra becomes

$$\left[ \triangleright_{\bar{m}_1}(\sigma), \triangleright_{\bar{m}_2 \bar{m}_3}(\sigma') \right] = i \epsilon_{\bar{m}_1 \bar{m}_2 \bar{m}_3} \partial_\sigma \delta(\sigma - \sigma'). \tag{26}$$

Table 4: Constraints, dimensional reduction conditions and section conditions of $\mathcal{T}$-string.

| | Virasoro : $\mathcal{S}^{\underline{m}}$ | Gauß law : $\mathcal{U}_m$ |
|---|---|---|
| dimensional reduction | $p_{\underline{m_1 m_2}}(\sigma) \epsilon^{\underline{m_1} \cdots \underline{m_4}} \dfrac{\partial}{\partial x_0^{\underline{m_3 m_4}}}$ | none |
| section condition | $\dfrac{\partial}{\partial x_0^{\underline{m_1 m_2}}} \epsilon^{\underline{m_1} \cdots \underline{m_4}} \dfrac{\partial}{\partial x_0^{\underline{m_3 m_4}}}$ | none |

This is equivalent to the O(D,D) current algebra which is given by $\triangleright_M = (\triangleright_{\bar{m}}, \triangleright^{\bar{m}})$ with $\triangleright^{\bar{m}} = \frac{1}{2}\epsilon^{\bar{m}\bar{m}_1\bar{m}_2} \triangleright_{\bar{m}_1\bar{m}_2}$ and the O(D,D) invariant metric $\eta_{MN} = \epsilon_{\bar{m}_1\bar{m}_2\bar{m}_3}$ as

$$\left[ \triangleright_M(\sigma), \triangleright_N(\sigma') \right] = i\eta_{MN}\partial_\sigma\delta(\sigma - \sigma') . \tag{27}$$

The Virasoro operators become

$$\begin{cases} \mathcal{S} = p_{\bar{m}}\partial_\sigma x^{\bar{m}} = \frac{1}{2}\triangleright_M \eta^{MN} \triangleright_N , \\ \mathcal{H} = \frac{1}{2}p_{\bar{m}}\hat{\eta}^{\bar{m}\bar{n}}p_{\bar{n}} + \frac{1}{2}\partial_\sigma x^{\bar{m}}\hat{\eta}_{\bar{m}\bar{n}}\partial_\sigma x^{\bar{n}} = \frac{1}{2}\triangleright_M \hat{\eta}^{MN} \triangleright_N , \end{cases} \tag{28}$$

with the double Lorentz invariant metric $\hat{\eta}^{MN}$. There are no further conditions of the Virasoro and the Gauß law constraints; $\mathcal{S}^{\underline{m}} = 0 = \mathcal{U}_m$.

# 3 $\mathcal{A}$5-brane Lagrangians

## 3.1 $\mathcal{A}$5-brane Lagrangian with SL(5) U-duality symmetry

The SL(5) U-duality symmetry is manifestly realized by the $\mathcal{A}$5-brane. The spacetime background is described by the vielbein which is a SL(5)/SO(5) coset element $E_m{}^a$ satisfying

$$E_{m_1}{}^{a_1}E_{m_2}{}^{a_2}E_{m_3}{}^{a_3}E_{m_4}{}^{a_4}E_{m_5}{}^{a_5}\epsilon_{a_1a_2a_3a_4a_5} = \epsilon_{m_1m_2m_3m_4m_5} , \tag{29}$$

with $m, a = 1, \ldots, 5$. The background metrices with tensor indices are

$$\begin{aligned} G_{mn} &= E_m{}^a\hat{\eta}_{ab}E_n{}^b , \\ G_{m_1m_2;n_1n_2} &= E_{m_1}{}^{a_1}E_{m_2}{}^{a_2}\hat{\eta}_{a_1[b_1}\hat{\eta}_{b_2]a_2}E_{n_1}{}^{b_1}E_{n_2}{}^{b_2} . \end{aligned} \tag{30}$$

The selfdual and anti-selfdual currents in a flat background $\mathring{F}_{\mathrm{SD}/\overline{\mathrm{SD}}}{}^{mn}$ and in a curved background $F_{\mathrm{SD}/\overline{\mathrm{SD}}}{}^{ab}$ in terms of (5) are given as

$$\begin{cases} \mathring{F}_{\mathrm{SD}}{}^{m_1m_2} = F_\tau{}^{m_1m_2} - \frac{1}{2}\epsilon^{m_1\cdots m_5}s_{m_3}F_{\sigma;m_4m_5} + g\hat{\eta}^{m_1n_1}\hat{\eta}^{m_2n_2}F_{\sigma;n_1n_2} , \\ \mathring{F}_{\overline{\mathrm{SD}}}{}^{m_1m_2} = F_\tau{}^{m_1m_2} - \frac{1}{2}\epsilon^{m_1\cdots m_5}s_{m_3}F_{\sigma;m_4m_5} - g\hat{\eta}^{m_1n_1}\hat{\eta}^{m_2n_2}F_{\sigma;n_1n_2} , \end{cases} \tag{31}$$

$$F_{\mathrm{SD}/\overline{\mathrm{SD}}}{}^{a_1a_2} = \mathring{F}_{\mathrm{SD}/\overline{\mathrm{SD}}}{}^{m_1m_2}E_{m_1}{}^{a_1}E_{m_2}{}^{a_2} , \tag{32}$$

where $\hat{\eta}^{mn}$ becomes $G^{mn}$ in a curved background. $g$ and $s_m$ are 5-brane world-volume vielbein fields which are introduced as Lagrange multipliers of Virasoro constraints.

The Lagrangian of the $\mathcal{A}$5-brane $L_{\mathrm{SL}(5)}$ is given [1] as

$$\begin{aligned} I_{\mathrm{SL}(5)} &= \int d\tau d^5\sigma \, L_{\mathrm{SL}(5)} , \\ L_{\mathrm{SL}(5)} &= \frac{1}{2}\phi F_{\mathrm{SD}}{}^{ab}F_{\overline{\mathrm{SD}}ab} + \frac{1}{2}\bar{\phi}(F_{\overline{\mathrm{SD}}}{}^{ab})^2 + \frac{1}{2}\lambda_{ab}F_{\overline{\mathrm{SD}}}{}^{ac}F_{\overline{\mathrm{SD}}}{}^b{}_c - \frac{1}{4}\epsilon_{a_1\cdots a_5}\lambda^{a_1}F_{\overline{\mathrm{SD}}}{}^{a_2a_3}F_{\overline{\mathrm{SD}}}{}^{a_4a_5} \\ &= \frac{\phi}{4}\mathring{F}_{\mathrm{SD}}{}^{m_1m_2}G_{m_1m_2;m_3m_4}\mathring{F}_{\overline{\mathrm{SD}}}{}^{m_3m_4} + \frac{\bar{\phi}}{8}\mathring{F}_{\overline{\mathrm{SD}}}{}^{m_1m_2}G_{m_1m_2;m_3m_4}\mathring{F}_{\overline{\mathrm{SD}}}{}^{m_3m_4} \\ &\quad + \frac{1}{2}\mathring{\lambda}_{mn}F_{\overline{\mathrm{SD}}}{}^{ml_1}G_{l_1l_2}\mathring{F}_{\overline{\mathrm{SD}}}{}^{nl_2} + \frac{1}{8}\epsilon_{m_1\cdots m_5}\mathring{\lambda}^{m_1}\mathring{F}_{\overline{\mathrm{SD}}}{}^{m_2m_3}\mathring{F}_{\overline{\mathrm{SD}}}{}^{m_4m_5} , \end{aligned} \tag{33}$$

with symmetric traceless tensors $\lambda_{\hat{m}\hat{n}}$'s.

## 3.2 World-volume covariant $\mathcal{A}$5-brane Lagrangian with SL(6) duality symmetry

The $G$=SL(5) U-duality symmetry is enlarged to $A$=SL(6) by cooperating with the 6-dimensional world-volume Lorentz covariance. The SL(6)/SO(6) coset parameter includes not only the target space vielbein SL(5)/SO(5) but also 6 components of the world-volume vielbein. The background vielbein $E_{\hat{m}}{}^{\hat{a}} \in$ SL(6)/SO(6) satisfies

$$E_{\hat{m}_1}{}^{\hat{a}_1} E_{\hat{m}_2}{}^{\hat{a}_2} E_{\hat{m}_3}{}^{\hat{a}_3} E_{\hat{m}_4}{}^{\hat{a}_4} E_{\hat{m}_5}{}^{\hat{a}_5} E_{\hat{m}_6}{}^{\hat{a}_6} \epsilon_{\hat{a}_1\hat{a}_2\hat{a}_3\hat{a}_4\hat{a}_5\hat{a}_6} = \epsilon_{\hat{m}_1\hat{m}_2\hat{m}_3\hat{m}_4\hat{m}_5\hat{m}_6}\,, \tag{34}$$

with $\hat{m}, \hat{a} = 0, 1, \ldots, 5$.

This SL(6) covariant vielbein (34) includes the 5-brane world-volume vielbein fields $g$ and $s_m$ as

$$E_{\hat{m}}{}^{\hat{a}} = \begin{pmatrix} E_0{}^{\hat{0}} & E_0{}^a \\ E_m{}^{\hat{0}} & E_m{}^a \end{pmatrix} = \begin{pmatrix} \dfrac{1}{g} & 0 \\ -\dfrac{s_m}{g} & E_m{}^a \end{pmatrix}\,, \tag{35}$$

with $\hat{m} = (0, m)$, $\hat{a} = (\hat{0}, a)$ and $m, a = 1 \cdots, 5$. It is denoted that the $E_m{}^a$ component of SL(6) vielbein (34) is different from the SL(5) vielbein $E_m{}^a$ in (35) up to the determinant factor. The number of degrees of freedom of the SL(6) vielbein is sum of the spacetime vielbein and the world-volume vielbein as

$$(6^2 - 1) - \frac{6 \times 5}{2} = \left((5^2 - 1) - \frac{5 \times 4}{2}\right) + 6\,. \tag{36}$$

This is generalized for a $p$-brane of $\mathcal{A}$-theory symmetry with $A/L$ coset as

$$\dim \frac{A}{L} = \dim \frac{G}{H} + (p + 1)\,. \tag{37}$$

The SL(6) covariant field strengths are given by a simple form; the one in a flat background $\mathring{F}^{\hat{m}\hat{n}\hat{l}}$ ( the same as (4) ) and the one in a curved background $F^{\hat{a}\hat{b}\hat{c}}$ as

$$\mathring{F}^{\hat{m}\hat{n}\hat{l}} = \frac{1}{2}\partial^{[\hat{m}}X^{\hat{n}\hat{l}]}\,, \qquad F^{\hat{a}\hat{b}\hat{c}} = \mathring{F}^{\hat{m}\hat{n}\hat{l}} E_{\hat{m}}{}^{\hat{a}} E_{\hat{n}}{}^{\hat{b}} E_{\hat{l}}{}^{\hat{c}}\,. \tag{38}$$

The selfdual and the anti-selfdual field strength (31) and (32) are written in terms of the SL(6) current (38) with $\partial^{\hat{0}} = \partial_\tau$ and $\epsilon^{\hat{0}12345} = 1$ as

$$F_{\text{SD}/\overline{\text{SD}}}{}^{a_1 a_2} = g \left( F^{\hat{0}a_1a_2} \pm \frac{1}{6}\epsilon^{\hat{0}a_1a_2}{}_{a_3a_4a_5} F^{a_3a_4a_5}\right)\,. \tag{39}$$

Then the $\mathcal{A}$5-brane Lagrangian (33) is rewritten in terms of the SL(6) covariant field strength (38). The world-volume covariant $\mathcal{A}$5-brane Lagrangian $L_{\text{SL}(6)}$ is given [1] as

$$\begin{aligned}
I_{\text{SL}(6)} &= \int d^6\sigma \; L_{\text{SL}(6)}\,, \\
L_{\text{SL}(6)} &= -\frac{1}{12}\Phi F^{\hat{a}_1\hat{a}_2\hat{a}_3} F_{\hat{a}_1\hat{a}_2\hat{a}_3} + \frac{1}{2}\Lambda_{\hat{a}\hat{b}} F^{\hat{a}\hat{c}_1\hat{c}_2} F^{\hat{b}}{}_{\hat{c}_1\hat{c}_2} + \frac{1}{12}\epsilon_{\hat{a}_1\cdots\hat{a}_6}\tilde{\Lambda}_{\hat{b}}{}^{\hat{a}_1} F^{\hat{a}_2\hat{a}_3\hat{a}_4} F^{\hat{a}_5\hat{a}_6\hat{b}} \\
&= -\frac{1}{72}\Phi\mathring{F}^{\hat{m}_1\hat{m}_2\hat{m}_3} G_{\hat{m}_1\hat{m}_2\hat{m}_3;\hat{m}_4\hat{m}_5\hat{m}_6}\mathring{F}^{\hat{m}_4\hat{m}_5\hat{m}_6} + \frac{1}{8}\mathring{\Lambda}_{\hat{m}\hat{n}}\mathring{F}^{\hat{m}\hat{l}_1\hat{l}_2} G_{\hat{l}_1\hat{l}_2;\hat{l}_3\hat{l}_4}\mathring{F}^{\hat{n}\hat{l}_3\hat{l}_4} \\
&\quad + \frac{1}{12}\epsilon_{\hat{m}_1\cdots\hat{m}_6}\mathring{\tilde{\Lambda}}_{\hat{n}}{}^{\hat{m}_1}\mathring{F}^{\hat{m}_2\hat{m}_3\hat{m}_4}\mathring{F}^{\hat{m}_5\hat{m}_6\hat{n}}\,,
\end{aligned} \tag{40}$$

where $\Phi$, $\Lambda_{\hat{a}\hat{b}}$ are Lagrange multipliers with symmetric traceless tensors $\Lambda_{\hat{a}\hat{b}}$'s. The background metrices with tensor indices are

$$\begin{aligned}
G_{\hat{m}_1\hat{m}_2;\hat{n}_1\hat{n}_2} &= E_{\hat{m}_1}{}^{\hat{a}_1} E_{\hat{m}_2}{}^{\hat{a}_2} \hat{\eta}_{\hat{a}_1[\hat{b}_1}\hat{\eta}_{\hat{b}_2]\hat{a}_2} E_{\hat{n}_1}{}^{\hat{b}_1} E_{\hat{n}_2}{}^{\hat{b}_2}\,, \\
G_{\hat{m}_1\hat{m}_2\hat{m}_3;\hat{n}_1\hat{n}_2\hat{n}_3} &= E_{\hat{m}_1}{}^{\hat{a}_1} E_{\hat{m}_2}{}^{\hat{a}_2} E_{\hat{m}_3}{}^{\hat{a}_3} \hat{\eta}_{\hat{a}_1[\hat{b}_1}\hat{\eta}_{\hat{b}_2|\hat{a}_2|}\hat{\eta}_{\hat{b}_3]\hat{a}_3} E_{\hat{n}_1}{}^{\hat{b}_1} E_{\hat{n}_2}{}^{\hat{b}_2} E_{\hat{n}_3}{}^{\hat{b}_3}\,.
\end{aligned} \tag{41}$$

# 4 Lagrangian of D-dimensional string from O(D,D) $\mathcal{T}$-string

In this section we derive the O(D,D) $\mathcal{T}$-string Lagrangian from the O(D,D) Hamiltonian by the double zweibein method [58, 59]. Then the reduction procedure from the O(D,D) $\mathcal{T}$-string Lagrangian to the conventional string Lagrangian is presented.

## 4.1 O(D,D) $\mathcal{T}$-string

We begin with the sigma model string Lagrangian

$$
\begin{aligned}
I &= \int d^2\sigma \, L \,, \\
L &= -\frac{1}{2}\partial_\mu x^m (\sqrt{-h}h^{\mu\nu}g_{mn} + \epsilon^{\mu\nu}B_{mn})\partial_\nu x^n \,,
\end{aligned}
\tag{42}
$$

with $\mu = (\tau, \sigma)$. In the conformal gauge the Lagrangian becomes

$$
\begin{aligned}
L &= \frac{1}{2}(\dot{x}^m g_{mn}\dot{x}^n - x'^m g_{mn}x'^n) - \dot{x}^m B_{mn}x'^n \\
&= \frac{1}{2}(\dot{x}^m \; x'^m)\begin{pmatrix} g_{mn} & B_{mn} \\ B_{mn} & g_{mn} \end{pmatrix}\begin{pmatrix} 1 & 0 \\ 0 & -1 \end{pmatrix}\begin{pmatrix} \dot{x}^n \\ x'^n \end{pmatrix} \\
&= \frac{1}{2}\partial_+ x^m (g_{mn} + B_{mn})\partial_- x^n \,,
\end{aligned}
\tag{43}
$$

with $\dot{x} = \partial_\tau x$, $x' = \partial_\sigma x$ and $\partial_\pm x = \dot{x} \pm x'$.

The Hamiltonian is given by the Legendre transformation where the canonical momentum of $x^m$ is given by $p_m = \partial L/\partial \dot{x}^m$,

$$
\begin{aligned}
H &= p_m \dot{x}^m - L \\
&= \frac{1}{2}(p_m \; x'^m)\begin{pmatrix} g^{mn} & g^{ml}B_{ln} \\ -B_{ml}g^{ln} & g_{mn} - B_{ml}g^{lk}B_{kn} \end{pmatrix}\begin{pmatrix} p_n \\ x'^n \end{pmatrix} \\
&= \frac{1}{2}\{(p_m - x'^l B_{lm})g^{mn}(p_n + B_{nk}x'^k) + x'^m g_{mn}x'^n\} \,.
\end{aligned}
\tag{44}
$$

The background field is the O(D,D) matrix $G^{MN}$ written in terms of the vielbein $E_A{}^M$ as $E_A{}^M \to h_A{}^B E_B{}^N g_N{}^M$, $h \in$SO(D−1,1) and $g \in$ O(D,D)

$$
E_A{}^M \eta_{MN} E_B{}^N = \eta_{AB} \,.
\tag{45}
$$

The background metric $G^{MN}$ in the string Hamiltonian (44) is given as

$$
\begin{aligned}
G^{MN} &= \begin{pmatrix} g^{mn} & g^{ml}B_{ln} \\ -B_{ml}g^{ln} & g_{mn} - B_{ml}g^{lk}B_{kn} \end{pmatrix} = E_A{}^M \hat{\eta}^{AB} E_B{}^N \,, \\
E_A{}^M &= \begin{pmatrix} e_a{}^m & e_a{}^l B_{lm} \\ 0 & e_m{}^a \end{pmatrix} \,,
\end{aligned}
\tag{46}
$$

while its inverse is given by

$$
\begin{aligned}
G_{MN} &= \begin{pmatrix} g_{mn} - B_{ml}g^{lk}B_{kn} & -B_{ml}g^{ln} \\ g^{ml}B_{ln} & g^{mn} \end{pmatrix} = E_M{}^A \hat{\eta}_{AB} E_N{}^B \,, \\
E_M{}^A &= \begin{pmatrix} e_m{}^a & -B_{ml}e_a{}^l \\ 0 & e_a{}^m \end{pmatrix} \,.
\end{aligned}
\tag{47}
$$

This O(D,D) background metric is utilized in the Lagrangian with manifest O(D,D) T-duality symmetry.

The O(D,D) covariant space is constructed in such a way that the O(D,D) covariant derivative $\triangleright_M(\sigma)$ algebra satisfies the same algebra of $\triangleright_M = (p_m, x'^m)$ up to the normalization

$$[\triangleright_M(\sigma), \triangleright_N(\sigma')] = 2i\eta_{MN}\partial_\sigma\delta(\sigma-\sigma'), \qquad \eta_{MN} = \begin{pmatrix} & \delta^n_m \\ \delta^m_n & \end{pmatrix}. \tag{48}$$

The covariant derivative $\triangleright_M$ is realized in terms of the doubled coordinate $X^M$ and $P_M$ with $[P_M(\sigma), X^N(\sigma')] = -i\delta^N_M\delta(\sigma-\sigma')$ as

$$\overset{\circ}{\triangleright}_M = P_M + \partial_\sigma X^N \eta_{NM}, \tag{49}$$

which is left moving current in the doubled space. The right moving current is also introduced as

$$\tilde{\triangleright}_M = P_M - \partial_\sigma X^N \eta_{NM}, \tag{50}$$

which satisfies the same current algebra (48) with opposite sign. The number of canonical variables of the doubled space are 4D, while the physical one is 2D. The 2D equations $\tilde{\triangleright}_M = 0$ is the usual selfduality condition to suppress 2D unphysical degrees of freedom, so we call $\tilde{\triangleright}_M$ "anti-selfdual current". Another current $\overset{\circ}{\triangleright}_M$ is selfdual current.

There are two sets of Virasoro operatros written in terms of the selfdual current and the anti-selfdual current

$$\begin{cases} \mathcal{H} = \frac{1}{4}\overset{\circ}{\triangleright}_M \hat{\eta}^{MN} \overset{\circ}{\triangleright}_N, \\ \mathcal{S} = \frac{1}{4}\overset{\circ}{\triangleright}_M \eta^{MN} \overset{\circ}{\triangleright}_N, \end{cases} \qquad \begin{cases} \tilde{\mathcal{H}} = \frac{1}{4}\tilde{\triangleright}_M \hat{\eta}^{MN} \tilde{\triangleright}_N, \\ \tilde{\mathcal{S}} = \frac{1}{4}\tilde{\triangleright}_M \eta^{MN} \tilde{\triangleright}_N. \end{cases} \tag{51}$$

$\mathcal{H}$ and $\mathcal{S}$ satisfy the Virasoro algebra

$$\begin{aligned} [\mathcal{S}(\sigma), \mathcal{S}(\sigma')\}] &= i\{\mathcal{S}(\sigma)+\mathcal{S}(\sigma')\}\partial_\sigma\delta(\sigma-\sigma'), \\ [\mathcal{S}(\sigma), \mathcal{H}(\sigma')\}] &= i\{\mathcal{H}(\sigma)+\mathcal{H}(\sigma')\}\partial_\sigma\delta(\sigma-\sigma'), \\ [\mathcal{H}(\sigma), \mathcal{H}(\sigma')\}] &= i\{\mathcal{S}(\sigma)+\mathcal{S}(\sigma')\}\partial_\sigma\delta(\sigma-\sigma'), \end{aligned} \tag{52}$$

while $\tilde{\mathcal{H}}$ and $\tilde{\mathcal{S}}$ satisfy the same Virasoro algebra with opposite signs on the right hand side.

As seen in the Hamiltonian in curved background (44) currents $\triangleright_M$ coupled to the vielbein as

$$\triangleright_A = E_A{}^M \overset{\circ}{\triangleright}_M, \quad \tilde{\triangleright}_A = E_A{}^M \tilde{\triangleright}_M. \tag{53}$$

In curved background the Virasoro constraints become

$$\begin{cases} \mathcal{H} = \frac{1}{4}\triangleright_A \hat{\eta}^{AB} \triangleright_B = \frac{1}{4}\overset{\circ}{\triangleright}_M G^{MN} \overset{\circ}{\triangleright}_N, \\ \mathcal{S} = \frac{1}{4}\triangleright_A \eta^{AB} \triangleright_B = \frac{1}{4}\overset{\circ}{\triangleright}_M \eta^{MN} \overset{\circ}{\triangleright}_N, \end{cases} \qquad \begin{cases} \tilde{\mathcal{H}} = \frac{1}{4}\tilde{\triangleright}_A \hat{\eta}^{AB} \tilde{\triangleright}_B = \frac{1}{4}\tilde{\triangleright}_M G^{MN} \tilde{\triangleright}_N, \\ \tilde{\mathcal{S}} = \frac{1}{4}\tilde{\triangleright}_A \eta^{AB} \tilde{\triangleright}_B = \frac{1}{4}\tilde{\triangleright}_M \eta^{MN} \tilde{\triangleright}_N. \end{cases} \tag{54}$$

The O(D,D) covariant Hamiltonian is given by the sum of all these Virasoro constraints with Lagrange multipliers which are doubled zweibeins [58]

$$\begin{aligned} H &= g\mathcal{H} + s\mathcal{S} + \tilde{g}\tilde{\mathcal{H}} + \tilde{s}\tilde{\mathcal{S}} \\ &= \frac{1}{2}\left[P_A M^{AB} P_B + 2P_A N^{AC}\eta_{CB}X'^B + X'^A \eta_{AC} M^{CD}\eta_{DB}X'^B\right], \end{aligned} \tag{55}$$

with $P_A = P_M E_A{}^M$ and $X'^A \equiv X'^M E_M{}^A$. We used the fact that the covariant derivatives are rewritten as $\triangleright_A = P_A + X'^B\eta_{BA}$ and $\tilde{\triangleright}_A = P_A - X'^B\eta_{BA}$ by the orthogonal condition (45). Matrices $M^{AB}$ and $N^{AB}$ are given as

$$\begin{aligned} M^{AB} &= \frac{g+\tilde{g}}{2}\hat{\eta}^{AB} + \frac{s+\tilde{s}}{2}\eta^{AB}, \\ N^{AB} &= \frac{g-\tilde{g}}{2}\hat{\eta}^{AB} + \frac{s-\tilde{s}}{2}\eta^{AB}, \end{aligned} \tag{56}$$

with the inverse of $M^{AB}$ as

$$M^{-1}{}_{AB} = \frac{2}{(g+\tilde{g})^2 - (s+\tilde{s})^2} \{(g+\tilde{g})\hat{\eta}_{AB} - (s+\tilde{s})\eta_{AB}\} \,. \tag{57}$$

The Legendre transformation of the Hamiltonian (55) with (56) leads to the following Lagrangian

$$L = P_M \dot{X}^M - H \;=\; \frac{1}{2} J_+{}^A M^{-1}{}_{AB} J_-{}^B \,, \tag{58}$$

$$\begin{cases} J_+{}^A = \dot{X}^A + (\tilde{g}\hat{\eta}^{AB} + \tilde{s}\eta^{AB})\eta_{BC} X'^C \,, \\ J_-{}^A = \dot{X}^A - (g\hat{\eta}^{AB} + s\eta^{AB})\eta_{BC} X'^C \,, \end{cases} \tag{59}$$

with $\dot{X}^A \equiv \dot{X}^M E_M{}^A$.

The Lagrangian in (58) can be written in terms of the selfdual current and the anti-selfdual current which is equal to $J_-$ in (59). The selfdual and anti-selfdual currents are given by

$$\begin{cases} J_{\mathrm{SD}}{}^A = (\dot{X}^A - sX'^A) + g\hat{\eta}^{AB}\eta_{BC} X'^C \,, \\ J_{\overline{\mathrm{SD}}}{}^A = (\dot{X}^A - sX'^A) - g\hat{\eta}^{AB}\eta_{BC} X'^C \,. \end{cases} \tag{60}$$

The selfdual and anti-selfdual currents in the flat background, $J_{\mathrm{SD}/\overline{\mathrm{SD}}}{}^M = J_{\mathrm{SD}/\overline{\mathrm{SD}}}{}^A E_A{}^M$, are written as

$$\begin{cases} \mathring{J}_{\mathrm{SD}}{}^M = (\dot{X}^M - sX'^M) + g\hat{\eta}^{MN}\eta_{NL} X'^L \,, \\ \mathring{J}_{\overline{\mathrm{SD}}}{}^M = (\dot{X}^M - sX'^M) - g\hat{\eta}^{MN}\eta_{NL} X'^L \,. \end{cases} \tag{61}$$

It is denoted that $\hat{\eta}^{MN}$ becomes $G^{MN}$ in a curved background. The resultant O(D,D) covariant Lagrangian for a $\mathcal{T}$-string is given [58] as

$$\begin{aligned} I &= \int d\tau d\sigma \; L \,, \\ L &= \phi J_{\mathrm{SD}}{}^A \hat{\eta}_{AB} J_{\overline{\mathrm{SD}}}{}^B + \bar{\phi} J_{\overline{\mathrm{SD}}}{}^A \hat{\eta}_{AB} J_{\overline{\mathrm{SD}}}{}^B + \tilde{\phi} J_{\overline{\mathrm{SD}}}{}^A \eta_{AB} J_{\overline{\mathrm{SD}}}{}^B \\ &= \phi \mathring{J}_{\mathrm{SD}}{}^M G_{MN} \mathring{J}_{\overline{\mathrm{SD}}}{}^N + \bar{\phi} \mathring{J}_{\overline{\mathrm{SD}}}{}^M G_{MN} \mathring{J}_{\overline{\mathrm{SD}}}{}^N + \tilde{\phi} \mathring{J}_{\overline{\mathrm{SD}}}{}^M \eta_{MN} \mathring{J}_{\overline{\mathrm{SD}}}{}^N \,. \end{aligned} \tag{62}$$

The first term is the kinetic term, while the rest are constraints that are squares of the anti-selfdual currents. The Lagrange multipliers $\phi$, $\bar{\phi}$ and $\tilde{\phi}$ are related to the doubled zweibeins as

$$\begin{cases} \phi = \dfrac{1}{2g} \,, \\ \bar{\phi} = \dfrac{1}{2g[(g+\tilde{g})^2 - (s+\tilde{s})^2]} \left\{ (s+\tilde{s})^2 + g^2 - \tilde{g}^2 \right\} \,, \\ \tilde{\phi} = -\dfrac{s+\tilde{s}}{(g+\tilde{g})^2 - (s+\tilde{s})^2} \,. \end{cases} \tag{63}$$

## 4.2 String from O(D,D) $\mathcal{T}$-string

We break the O(D,D) T-duality symmetry of $\mathcal{T}$-string into the GL(D) symmetry of the usual string. The background gauge field of $\mathcal{T}$-string is O(D,D)/O(D−1,1)$^2$ coset parameter which includes the D-dimensional metric $g_{mn}$ and $B_{mn}$ field, while the background gauge field of a string is GL(D)/SO(D-1,1) coset parameter which includes only $g_{mn}$. In this subsection we use the coordinate $X^M = (x^m, y_m)$ with off-diagonal $\eta_{MN}$ to describe $\mathcal{T}$-string, while the left/right moving coordinate with diagonal $\eta_{MN} = (\mathbf{1}, -\mathbf{1})$ was used in the reference [58]. The Weyl/Lorentz gauge of the zweibein [58] is given as

$$\varepsilon_\pm{}^\mu = \begin{pmatrix} \varepsilon_+{}^\tau & \varepsilon_+{}^\sigma \\ \varepsilon_-{}^\tau & \varepsilon_-{}^\sigma \end{pmatrix} = \begin{pmatrix} 1 & g-s \\ 1 & -g-s \end{pmatrix} \,. \tag{64}$$

The left/right moving modes with the zweibein is $\varepsilon_{\pm} X \equiv \varepsilon_{\pm}{}^{\mu} \partial_{\mu} X$. The selfdual and anti-selfdual currents (61) are expressed as

$$
\begin{cases}
\overset{\circ}{J}_{\mathrm{SD}/\overline{\mathrm{SD}}}{}^{m} = \dot{x}^{m} - s x'^{m} \pm g \hat{\eta}^{mn} y'_{n} = \varepsilon_{\tau} x^{m} \pm g \hat{\eta}^{mn} \varepsilon_{\sigma} y_{n}\,, \\
\overset{\circ}{J}_{\mathrm{SD}/\overline{\mathrm{SD}};m} = \dot{y}_{m} - s y'_{m} \pm g \hat{\eta}_{mn} x'^{n} = \varepsilon_{\tau} y_{m} \pm g \hat{\eta}_{mn} \varepsilon_{\sigma} x^{n}\,,
\end{cases}
\tag{65}
$$

with

$$
\varepsilon_{\mu} X \equiv \varepsilon_{\mu}{}^{\nu} \partial_{\nu} X\,, \qquad \varepsilon_{\mu}{}^{\nu} = \begin{pmatrix} \varepsilon_{\tau}{}^{\tau} & \varepsilon_{\tau}{}^{\sigma} \\ \varepsilon_{\sigma}{}^{\tau} & \varepsilon_{\sigma}{}^{\sigma} \end{pmatrix} = \begin{pmatrix} 1 & -s \\ 0 & 1 \end{pmatrix}\,.
\tag{66}
$$

The condition of vanishing the anti-selfdual current in $s = 0$ and $g = 1$ gauge leads to the selfduality constraint in flat space as $\partial_{\mu} y = \epsilon_{\mu\nu} \partial^{\nu} x$ with $\partial^{\tau} = -\partial_{\tau}$. In the gauge $\phi = \frac{1}{2g}$ and $\bar{\phi} = 0 = \tilde{\phi}$, corresponding to $g = \tilde{g}$ and $s + \tilde{s} = 0$, the O(D,D) covariant Lagrangian (62) is written as

$$
\begin{aligned}
\frac{1}{2g} J_{\mathrm{SD}}{}^{A} \hat{\eta}_{AB} J_{\overline{\mathrm{SD}}}{}^{B} &= \frac{1}{2g} (\dot{X} - s X')^{A} \hat{\eta}_{AB} (\dot{X} - s X')^{B} - \frac{g}{2} X'^{C} \eta_{CD} \hat{\eta}^{DA} \hat{\eta}_{AB} \hat{\eta}^{BF} \eta_{FG} X'^{G} \\
&= \frac{1}{2g} (\dot{X} - s X')^{M} E_{M}{}^{A} \hat{\eta}_{AB} E_{N}{}^{B} (\dot{X} - s X')^{N} - \frac{g}{2} X'^{M} \eta_{ML} E_{A}{}^{L} \hat{\eta}^{AB} E_{B}{}^{K} \eta_{KN} X'^{N} \\
&= \frac{1}{2g} \varepsilon_{+} X^{M} G_{MN} \varepsilon_{-} X^{N} \\
&= \frac{1}{2g} \varepsilon_{+} X^{M} E_{M}{}^{A} \hat{\eta}_{AB} E_{N}{}^{B} \varepsilon_{-} X^{N}\,.
\end{aligned}
\tag{67}
$$

The orthogonality condition is used in the second equality, $\eta_{ML} G^{LK} \eta_{KN} = G_{MN}$, so $E_{M}{}^{A} \eta_{AB} = \eta_{MN} E_{B}{}^{N}$, is used in the last equality. In terms of $x^{m}, y_{m}$ coordinates it is given by

$$
\begin{aligned}
&\frac{1}{2g} J_{\mathrm{SD}}{}^{A} \hat{\eta}_{AB} J_{\overline{\mathrm{SD}}}{}^{B} \\
&= \frac{1}{2g} (\varepsilon_{+} x^{m}\, \varepsilon_{+} y_{m}) \begin{pmatrix} g_{mn} - B_{ml} g^{lk} B_{kn} & -B_{ml} g^{ln} \\ g^{ml} B_{ln} & g^{mn} \end{pmatrix} \begin{pmatrix} \varepsilon_{-} x^{n} \\ \varepsilon_{-} y_{n} \end{pmatrix} \\
&= \frac{1}{2g} (\varepsilon_{+} x^{m}\, \varepsilon_{+} y_{m}) \begin{pmatrix} e_{m}{}^{a} & -B_{ml} e_{a}{}^{l} \\ 0 & e_{a}{}^{m} \end{pmatrix} \begin{pmatrix} \eta_{ab} & 0 \\ 0 & \eta^{ab} \end{pmatrix} \begin{pmatrix} e_{n}{}^{b} & 0 \\ -B_{nk} e_{b}{}^{k} & e_{b}{}^{n} \end{pmatrix} \begin{pmatrix} \varepsilon_{-} x^{n} \\ \varepsilon_{-} y_{n} \end{pmatrix} \\
&= \frac{1}{2g} \left[ \varepsilon_{+} x^{m} g_{mn} \varepsilon_{-} x^{n} + (\varepsilon_{+} y_{m} - \varepsilon_{+} x^{l} B_{lm}) g^{mn} (\varepsilon_{-} y_{n} + B_{nk} \varepsilon_{-} x^{k}) \right]\,.
\end{aligned}
\tag{68}
$$

We break the O(D,D) symmetry into the GL(D) symmetry by the dimensional reduction (2). The resultant Lagrangian is the kinetic term of the usual string with the zweibein field;

$$
L_{0} = \frac{1}{2g}\, \varepsilon_{+} x^{m}\, g_{mn}\, \varepsilon_{-} x^{n}\,.
\tag{69}
$$

In order to obtain the Wess-Zumino term we add the total derivative term

$$
\partial_{\mu} (\epsilon^{\mu\nu} x^{m} \partial_{\nu} y_{m}) = \dot{x} y' - x' \dot{y} = -\frac{1}{2g} (\varepsilon_{+} x\, \varepsilon_{-} y - \varepsilon_{+} y\, \varepsilon_{-} x)\,,
\tag{70}
$$

to the O(D,D) Lagrangian $L$ (68)

$$
\begin{aligned}
&\frac{1}{2g} J_{\mathrm{SD}}{}^{A} \hat{\eta}_{AB} J_{\overline{\mathrm{SD}}}{}^{B} - \partial_{\mu} (\epsilon^{\mu\nu} x^{m} \partial_{\nu} y_{m}) \\
&= \frac{1}{2g} \big\{ \varepsilon_{+} x^{m} g_{mn} \varepsilon_{-}^{n} x + (\varepsilon_{+} y_{m} - \varepsilon_{+} x^{l} B_{lm} - \varepsilon_{+} x^{l} g_{lm}) g^{mn} (\varepsilon_{-} y_{n} + B_{nk} \varepsilon_{-} x^{k} + g_{nk} \varepsilon_{-} x^{k}) \\
&\qquad + \varepsilon_{+} x^{m} g_{mn} \varepsilon_{-} x^{n} + 2 \varepsilon_{+} x^{m} B_{mn} \varepsilon_{-} x^{n} \big\}\,.
\end{aligned}
\tag{71}
$$

By the dimensional reduction (2) the Lagrangian with the total derivative term reduces into the string Lagrangian in curved background with the Wess-Zumino term as the curved world-sheet version of (43),

$$L_0 + L_{\mathrm{WZ}} = \frac{1}{g}\, \varepsilon_+ x^m (g_{mn} + B_{mn}) \varepsilon_- x^n \,. \tag{72}$$

The zweibeins in (72) and (42) are related as

$$g = -\frac{2}{\sqrt{-h}h^{00}}\,, \qquad s = -\frac{h^{01}}{h^{00}}\,. \tag{73}$$

# 5 Lagrangians of string via $\mathcal{T}$-string from $\mathcal{A}5$-brane

In this section, we derive the $\mathcal{T}$-string Lagrangian from the $\mathcal{A}5$-brane Lagrangian. The resulting $\mathcal{T}$-string Lagrangian is formulated in terms of an SL(4) rank-two antisymmetric tensor coordinate, which is coupled to the string background. We then present the reduction procedure from the $\mathcal{T}$-string Lagrangian to the conventional string Lagrangian.

## 5.1 $\mathcal{T}$-string from $\mathcal{A}5$-brane

The O(3,3) $\mathcal{T}$-string from $\mathcal{A}5$-brane is described by the SL(4) rank-two anti-symmetric tensor coordinate $X^{\underline{mn}} = (x^{\bar m},\ y^{\bar m \bar n})$ with $\underline{m} = 1, \cdots, 4$ and $\bar m = 1, 2, 3$ as listed in Tab. 1. The SL(6) rank-two tensor coordinate is decomposed as SL(6) $\rightarrow$ SL(5) $\rightarrow$ SL(4) as $X^{\hat m \hat n} = (X^{0n} = Y^n,\ X^{mn}) \rightarrow X^{mn} = (X^{5\underline n} = Y^{\underline m},\ X^{\underline{mn}}) \rightarrow X^{\underline{mn}} = (X^{4\bar m} = x^{\bar m},\ X^{\bar m \bar n} = y^{\bar m \bar n})$ with $\hat m = 0, 1, \cdots, 5$ and $m = 1, \cdots, 5$. The 6-dimensional world-volume derivative is reduced into the string world-sheet derivatives as $\partial^{\hat m} = (\partial^0 = \partial_\tau,\ \partial^5 = \partial_\sigma,\ \partial^{\underline m} = 0)$. The SL(6) field strength for the $\mathcal{T}$-string has the following components

$$\overset{\circ}{F}{}^{0\underline{mn}} = \partial_\tau X^{\underline{mn}}, \qquad \overset{\circ}{F}{}^{5\underline{mn}} = \partial_\sigma X^{\underline{mn}}, \qquad \overset{\circ}{F}{}^{05\underline m} = 0 = \overset{\circ}{F}{}^{\underline{mnl}}. \tag{74}$$

The SL(6) vielbein for the $\mathcal{T}$-string has a block diagonal form as

$$E_{\hat m}{}^{\hat a} = \begin{pmatrix} E_0{}^{\hat 0} & E_0{}^{\hat 5} & E_0{}^{\underline a} \\ E_5{}^{\hat 0} & E_5{}^{\hat 5} & E_5{}^{\underline a} \\ E_{\underline m}{}^{\hat 0} & E_{\underline m}{}^{\hat 5} & E_{\underline m}{}^{\underline a} \end{pmatrix} = \begin{pmatrix} \dfrac{1}{g} & 0 & 0 \\ -\dfrac{s}{g} & 1 & 0 \\ 0 & 0 & g^{1/4} E_{\underline m}{}^{\underline a} \end{pmatrix}. \tag{75}$$

The selfdual and the anti-selfdual currents are the following combinations of the SL(6) field strengths in (74) with (38) as

$$J_{\mathrm{SD}/\overline{\mathrm{SD}}}{}^{\underline{a_1 a_2}} = g\left( F^{\hat 0 \underline{a_1 a_2}} \pm \frac{1}{2} \epsilon^{\hat 0 \underline{a_1 a_2}}{}_{5 \underline{a_3 a_4}} F^{\hat 5 \underline{a_3 a_4}} \right). \tag{76}$$

The zweibein fields $g$ and $s$ are part of the SL(6) vielbein (75) in the new SL(6) duality symmetry formulation in (38), contrast to that the world-volume vielbein fields are separated from the SL(4) spacetime vielbein $E_{\underline m}{}^{\underline a}$ in the SL(5) formulation in (31) and (32) as

$$J_{\mathrm{SD}/\overline{\mathrm{SD}}}{}^{\underline{a_1 a_2}} = \overset{\circ}{J}_{\mathrm{SD}/\overline{\mathrm{SD}}}{}^{\underline{m_1 m_2}} E_{\underline{m_1}}{}^{\underline{a_1}} E_{\underline{m_2}}{}^{\underline{a_2}}\,,$$
$$\overset{\circ}{J}_{\mathrm{SD}/\overline{\mathrm{SD}}}{}^{\underline{m_1 m_2}} = \varepsilon_\tau X^{\underline{m_1 m_2}} \pm \frac{1}{2}\hat\eta^{\underline{m_1 n_1}}\hat\eta^{\underline{m_2 n_2}}(-\epsilon_{\underline{n_1 \cdots n_4}})\varepsilon_\sigma X^{\underline{n_3 n_4}}\,, \tag{77}$$

with (64). The minus sign in the last equation is caused from $\epsilon_{\bar{m}_1\bar{m}_2\bar{m}_3}x^{\bar{m}_3} = -\epsilon_{\bar{m}_1\bar{m}_24\bar{m}_3}X^{4\bar{m}_3}$. The O(3,3) invariant metric $\eta_{MN}$ becomes SL(4) invariant metric $\epsilon_{\underline{m}_1\cdots\underline{m}_4}$. The current in (77) is written in terms of $x^{\bar{m}}$ and $y^{\bar{m}\bar{n}}$ as

$$
\begin{cases}
\mathring{J}_{\mathrm{SD}/\overline{\mathrm{SD}}}{}^{4\bar{m}} = \dot{x}^{\bar{m}} - sx'^{\bar{m}} \pm g\frac{1}{2}\epsilon^{\bar{m}}{}_{\bar{n}_1\bar{n}_2}y'^{\bar{n}_1\bar{n}_2} = \varepsilon_\tau x^{\hat{m}} \pm \frac{1}{2}\epsilon^{\hat{m}}{}_{\hat{n}_1\hat{n}_2}\varepsilon_\sigma y^{\hat{n}_1\hat{n}_2}, \\
\mathring{J}_{\mathrm{SD}/\overline{\mathrm{SD}}}{}^{\bar{m}_1\bar{m}_2} = \dot{y}^{\bar{m}_1\bar{m}_2} - sy'^{\bar{m}_1\bar{m}_2} \pm g\epsilon^{\bar{m}_1\bar{m}_2}{}_{\bar{n}}x'^{\bar{n}} = \varepsilon_\tau y^{\hat{m}_1\hat{m}_2} \pm \epsilon^{\hat{m}_1\hat{m}_2}{}_{\hat{n}}\varepsilon_\sigma x^{\hat{n}},
\end{cases}
\tag{78}
$$

which is related to the O(D,D) vector currents (65) with $y^{\bar{m}\bar{n}} \equiv \epsilon^{\bar{m}\bar{n}\bar{l}}y_{\bar{l}}$.

In order to obtain the usual 3-dimensional string Lagrangian we express the spacetime vielbein $E_{\underline{m}}{}^{\underline{a}} \in \mathrm{SL}(4)/\mathrm{SO}(4)$ in terms of the 3-dimensional metric $g_{\bar{m}\bar{n}}$ and the $B_{\bar{m}\bar{n}}$ field. The O(3,3) vector index contraction and the SL(4) tensor index contraction are assumed to be equal up to the normalization as

$$
dX^M E_M{}^A = dx^{\bar{m}}E_{\bar{m}}{}^A + dy_{\bar{m}}E^{\bar{m};A} = dX^{4\bar{x}}E_{4\bar{m}}{}^A + \frac{1}{2}dX^{\bar{m}\bar{n}}E_{\bar{m}\bar{n}}{}^A = \frac{1}{2}dX^{\underline{mn}}E_{\underline{mn}}{}^A.
\tag{79}
$$

We rewrite the O(D,D) vielbein in (47) in terms of tensor indices for D=3 case as

$$
\begin{aligned}
E_M{}^A &= \begin{pmatrix} e_{\bar{m}}{}^{\bar{a}} & -B_{\bar{m}\bar{l}}e_{\bar{b}}{}^{\bar{l}}\epsilon^{\bar{a}_1\bar{a}_2\bar{b}} \\ 0 & \epsilon_{\bar{m}_1\bar{m}_2\bar{m}}e_{\bar{b}}{}^{\bar{m}}\epsilon^{\bar{a}_1\bar{a}_2\bar{b}} \end{pmatrix} \\
&= c\begin{pmatrix} E_{4\bar{m}}{}^{4\bar{a}} & E_{4\bar{m}}{}^{\bar{a}_1\bar{a}_2} \\ E_{\bar{m}_1\bar{m}_2}{}^{4\bar{a}} & E_{\bar{m}_1\bar{m}_2}{}^{\bar{a}_1\bar{a}_2} \end{pmatrix} = cE_{\underline{m}_1\underline{m}_2}{}^{\underline{a}_1\underline{a}_2} = cE_{[\underline{m}_1}{}^{\underline{a}_1}E_{\underline{m}_2]}{}^{\underline{a}_2},
\end{aligned}
\tag{80}
$$

with a normalization factor $c$. The vielbein with the tensor indices can be written as the product of the one with the vector indices

$$
\begin{aligned}
E_{\underline{m}}{}^{\underline{a}} &= \begin{pmatrix} E_4{}^4 & E_4{}^{\bar{a}} \\ E_{\bar{m}}{}^4 & E_{\bar{m}}{}^{\bar{a}} \end{pmatrix} = \mathbf{e}^{-1/4}\begin{pmatrix} 1 & -\tilde{B}^{\bar{n}}e_{\bar{n}}{}^{\bar{a}} \\ 0 & e_{\bar{m}}{}^{\bar{a}} \end{pmatrix}, \\
\tilde{B}^{\bar{m}} &= \frac{1}{2}\epsilon^{\bar{m}\bar{n}\bar{l}}B_{\bar{n}\bar{l}}, \\
\mathbf{e} &= \det e_{\bar{m}}{}^{\bar{a}}.
\end{aligned}
\tag{81}
$$

The background gauge field in the tensor index is now

$$
\begin{aligned}
G_{MN} = G_{\underline{m}_1\underline{m}_2;\underline{n}_1\underline{n}_2} &= \frac{1}{2^2}E_{\underline{m}_1\underline{m}_2}{}^{\underline{a}_1\underline{a}_2}\hat{\eta}_{\underline{a}_1[\underline{b}_1}\hat{\eta}_{\underline{b}_2]\underline{a}_2}E_{\underline{n}_1\underline{n}_2}{}^{\underline{b}_1\underline{b}_2} \\
&= \begin{pmatrix} G_{\bar{m}\bar{n}} & G_{\bar{m};\bar{n}_1\bar{n}_2} \\ G_{\bar{m}_1\bar{m}_2;\bar{n}} & G_{\bar{m}_1\bar{m}_2;\bar{n}_1\bar{n}_2} \end{pmatrix} \\
&= \mathbf{e}^{-1}\begin{pmatrix} g_{\bar{m}\bar{n}} - \tilde{B}^{\bar{p}}g_{\bar{m}[\bar{p}}g_{\bar{n}]\bar{q}}\tilde{B}^{\bar{q}} & g_{\bar{m}[\bar{n}_1}g_{\bar{n}_2]\bar{l}}\tilde{B}^{\bar{l}} \\ -\tilde{B}^{\bar{p}}g_{\bar{p}[\bar{m}_1}g_{\bar{m}_2]\bar{n}} & g_{\bar{m}_1[\bar{n}_1}g_{\bar{n}_2]\bar{m}_2} \end{pmatrix},
\end{aligned}
\tag{82}
$$

where metric of the stability group is denoted as $\hat{\eta}_{\underline{mn}}$ to distinguish from $\eta_{MN}$.

The $\mathcal{T}$-string Lagrangian is obtained from the world-volume covariant $\mathcal{A}5$-Lagrangian (40)

$$
I = \int d^2\sigma \, L,
\tag{83}
$$

$$
\begin{aligned}
L = {} & \frac{\Phi}{2}\left(-(F^{\hat{0}\underline{a}_1\underline{a}_2})^2 + (F^{\hat{5}\underline{a}_1\underline{a}_2})^2\right) + \frac{1}{2}\Lambda_{\hat{0}\hat{5}}F^{\hat{0}\underline{a}_1\underline{a}_2}F^{\hat{5}}{}_{\underline{a}_1\underline{a}_2} + \frac{1}{2}\Lambda_{\underline{ab}}F^{\underline{a}\hat{0}\underline{c}}F^{\underline{b}}{}_{\hat{0}\underline{c}} + \frac{1}{2}\Lambda_{\underline{ab}}F^{\underline{a}\hat{5}\underline{c}}F^{\underline{b}}{}_{\hat{5}\underline{c}} \\
& + \frac{\epsilon_{\hat{0}\hat{5}\underline{a}_1\cdots\underline{a}_4}}{4}\left(\tilde{\Lambda}_{\hat{0}}{}^{\hat{0}}F^{\hat{5}\underline{a}_1\underline{a}_2}F^{\hat{0}\underline{a}_3\underline{a}_4} + \tilde{\Lambda}_{\hat{5}}{}^{\hat{0}}F^{\hat{5}\underline{a}_1\underline{a}_2}F^{\hat{5}\underline{a}_3\underline{a}_4} - \tilde{\Lambda}_{\hat{0}}{}^{\hat{5}}F^{\hat{0}\underline{a}_1\underline{a}_2}F^{\hat{0}\underline{a}_3\underline{a}_4} - \tilde{\Lambda}_{\hat{5}}{}^{\hat{5}}F^{\hat{0}\underline{a}_1\underline{a}_2}F^{\hat{5}\underline{a}_3\underline{a}_4}\right),
\end{aligned}
$$

with $\eta^{\hat{0}\hat{0}} = -1$ and $\eta^{\hat{5}\hat{5}} = 1$. Although currents are written as field strengths, there is no gauge symmetry of the coordinate $\delta X^{\underline{mn}}$. The $\mathcal{T}$-string Lagrangian in the SL(4) tensor coordinate is given by

$$
\begin{aligned}
L &= \phi \frac{1}{2^2} J_{\mathrm{SD}}{}^{\underline{a_1 a_2}} \hat{\eta}_{\underline{a_1}[\underline{b_1}} \hat{\eta}_{\underline{b_2}]\underline{a_2}} J_{\overline{\mathrm{SD}}}{}^{\underline{b_1 b_2}} + \bar{\phi} \frac{1}{2^2} J_{\overline{\mathrm{SD}}}{}^{\underline{a_1 a_2}} \hat{\eta}_{\underline{a_1}[\underline{b_1}} \hat{\eta}_{\underline{b_2}]\underline{a_2}} J_{\overline{\mathrm{SD}}}{}^{\underline{b_1 b_2}} \\
&\quad + \tilde{\phi} \frac{1}{2^2} J_{\overline{\mathrm{SD}}}{}^{\underline{a_1 a_2}} \epsilon_{\underline{a_1}\cdots\underline{a_4}} J_{\overline{\mathrm{SD}}}{}^{\underline{a_3 a_4}} \\
&= \phi \frac{1}{2^2} \mathring{J}_{\mathrm{SD}}{}^{\underline{m_1 m_2}} G_{\underline{m_1 m_2};\underline{n_1 n_2}} \mathring{J}_{\overline{\mathrm{SD}}}{}^{\underline{n_1 n_2}} + \bar{\phi} \frac{1}{2^2} \mathring{J}_{\overline{\mathrm{SD}}}{}^{\underline{m_1 m_2}} G_{\underline{m_1 m_2};\underline{n_1 n_2}} \mathring{J}_{\overline{\mathrm{SD}}}{}^{\underline{n_1 n_2}} \\
&\quad + \tilde{\phi} \frac{1}{2^2} \mathring{J}_{\overline{\mathrm{SD}}}{}^{\underline{m_1 m_2}} \epsilon_{\underline{m_1}\cdots\underline{m_4}} \mathring{J}_{\overline{\mathrm{SD}}}{}^{\underline{m_3 m_4}},
\end{aligned}
\tag{84}
$$

with the background metric $G_{\underline{m_1 m_2};\underline{n_1 n_2}}$ in (82).

The $\mathcal{T}$-string Lagrangian in the gauge $\Phi = g^2 \mathbf{e}$ and $\Lambda_{\hat{a}\hat{b}} = 0 = \tilde{\Lambda}_{\hat{a}}{}^{\hat{b}}$ as

$$
L = -\frac{g^2 \mathbf{e}}{2} \left( (F^{\hat{0}\underline{a_1 a_2}})^2 - (F^{\hat{5}\underline{a_1 a_2}})^2 \right).
\tag{85}
$$

The SL(4) covariant Lagrangian (84) in the gauge $\phi = \frac{1}{2g}$ and $\bar{\phi} = 0 = \tilde{\phi}$ is given as

$$
\begin{aligned}
L &= \frac{1}{2g} J_{\mathrm{SD}}{}^{A} \hat{\eta}_{AB} J_{\overline{\mathrm{SD}}}{}^{B} \\
&= \frac{1}{2^3 g} \varepsilon_+ X^{\underline{m_1 m_2}} G_{\underline{m_1 m_2};\underline{n_1 n_2}} \varepsilon_- X^{\underline{n_1 n_2}} \\
&= \frac{1}{2g} (\varepsilon_+ x^{\bar{m}} \; \varepsilon_+ y^{\bar{m}_1 \bar{m}_2}) \begin{pmatrix} g_{\bar{m}\bar{n}} - \tilde{B}^{\bar{p}} g_{\bar{m}[\bar{p}} g_{\bar{n}]\bar{q}} \tilde{B}^{\bar{q}} & g_{\bar{m}[\bar{n}_1} g_{\bar{n}_2]\bar{l}} \tilde{B}^{\bar{l}} \\ -\tilde{B}^{\bar{p}} g_{\bar{p}[\bar{m}_1} g_{\bar{m}_2]\bar{n}} & g_{\bar{m}_1[\bar{n}_1} g_{\bar{n}_2]\bar{m}_2} \end{pmatrix} \begin{pmatrix} \varepsilon_- x^{\bar{m}} \\ \varepsilon_- y^{\bar{m}_1 \bar{m}_2} \end{pmatrix}.
\end{aligned}
\tag{86}
$$

## 5.2 String from $\mathcal{T}$-string

We break SL(4) symmetry of $\mathcal{T}$-string into GL(3) for the usual string, where the reduction of the spacetime coordinate is performed as $X^{\underline{mn}} = (X^{4\bar{m}}, X^{\bar{m}\bar{n}}) = (x^{\bar{m}}, y^{\bar{m}\bar{n}}) \to x^{\bar{m}}$. We repeat the same procedure of subsection 4.2. The SL(4) Lagrangian (86) is rewritten analogously to (67)

$$
\begin{aligned}
\frac{1}{2g} J_{\mathrm{SD}}{}^{A} \hat{\eta}_{AB} J_{\overline{\mathrm{SD}}}{}^{B} &= \frac{1}{2g} \varepsilon_+ x^{\bar{m}} g_{\bar{m}\bar{n}} \varepsilon_- x^{\bar{n}} \\
&\quad + \frac{1}{2^3 g} \left( \varepsilon_+ y^{\bar{m}_1 \bar{m}_2} - \varepsilon_+ x^{[\bar{m}_1} \tilde{B}^{\bar{m}_2]} \right) g_{\bar{m}_1[\bar{n}_1} g_{\bar{n}_2]\bar{m}_2} \left( \varepsilon_- y^{\bar{n}_1 \bar{n}_2} + \tilde{B}^{[\bar{n}_1} \varepsilon_- x^{\bar{n}_2]} \right).
\end{aligned}
\tag{87}
$$

By the dimensional reduction (2) the Lagrangian (86) reduces to the kinetic term of the string (69).

The total derivative term which is added to obtain the Wess-Zumino term (70) becomes

$$
\begin{aligned}
-\frac{1}{2^2 g} \left( \varepsilon_+ x^{\bar{m}_1} \varepsilon_- y^{\bar{m}_2 \bar{m}_3} - \varepsilon_- x^{\bar{m}_1} \varepsilon_+ y^{\bar{m}_2 \bar{m}_3} \right) \epsilon_{\bar{m}_1 \bar{m}_2 \bar{m}_3} &= -\frac{1}{2} \partial_\mu \left( \epsilon^{\mu\nu} x^{\bar{m}_1} \partial_\nu y^{\bar{m}_2 \bar{m}_3} \epsilon_{\bar{m}_1 \bar{m}_2 \bar{m}_3} \right) \\
&= \frac{1}{2^2} \partial_\mu \left( \epsilon^{\mu\nu} X^{\underline{m_1 m_2}} \partial_\nu X^{\underline{m_3 m_4}} \epsilon_{\underline{m_1}\cdots\underline{m_4}} \right).
\end{aligned}
\tag{88}
$$

Adding this term to the SL(4) Lagrangian (86)

$$
\frac{1}{2g} J_{\mathrm{SD}}{}^{A} \hat{\eta}_{AB} J_{\overline{\mathrm{SD}}}{}^{B} + \frac{1}{2^2} \partial_\mu (\epsilon^{\mu\nu} X^{\underline{m_1 m_2}} \partial_\nu X^{\underline{m_3 m_4}} \epsilon_{\underline{m_1}\cdots\underline{m_4}})
\tag{89}
$$

$$
\begin{aligned}
&= \frac{1}{2g} \bigg\{ \varepsilon_+ x^{\bar{m}} g_{\bar{m}\bar{n}} \varepsilon_- x^{\bar{n}} + \frac{1}{2^2} (\varepsilon_+ y^{\bar{m}_1 \bar{m}_2} - \varepsilon_+ x^{[\bar{m}_1} \tilde{B}^{\bar{m}_2]} + \varepsilon_+ x^{\bar{l}_3} \epsilon_{\bar{l}_1 \bar{l}_2 \bar{l}_3} g^{\bar{l}_1 \bar{m}_1} g^{\bar{l}_2 \bar{m}_2}) g_{\bar{m}_1[\bar{n}_1} g_{\bar{n}_2]\bar{m}_2} \\
&\quad \times (\varepsilon_- y^{\bar{n}_1 \bar{n}_2} + \tilde{B}^{[\bar{n}_1} \varepsilon_- x^{\bar{n}_2]} - g^{\bar{n}_1 \bar{k}_1} g^{\bar{n}_2 \bar{k}_2} \epsilon_{\bar{k}_1 \bar{k}_2 \bar{k}_3} \varepsilon_- x^{\bar{k}_3}) + \varepsilon_+ x^{\bar{m}} g_{\bar{m}\bar{n}} \varepsilon_- x^{\bar{n}} + 2 \varepsilon_+ x^{\bar{m}} B_{\bar{m}\bar{n}} \varepsilon_- x^{\bar{n}} \bigg\}.
\end{aligned}
$$

After the dimensional reduction (2), the Lagrangian with the total derivative term reduces into the usual string Lagrangian with the Wess-Zumino term (72),

$$L_0 + L_{\text{WZ}} = \frac{1}{g}\, \varepsilon_+ x^{\bar{m}}(g_{\bar{m}\bar{n}} + B_{\bar{m}\bar{n}})\varepsilon_- x^{\bar{n}}\,.$$

# 6 Lagrangians of M2-brane via $\mathcal{M}$5-brane from $\mathcal{A}$5-brane

## 6.1 $\mathcal{M}$5-brane from $\mathcal{A}$5-brane

The GL(4) $\mathcal{M}$5-brane from $\mathcal{A}$5-brane is described by the GL(4) vector coordinate $X^{5\underline{m}} = x^{\underline{m}}$ [18] as listed in Tab. 1. The SL(6) rank-two tensor coordinate is decomposed as $SL(6) \to SL(5) \to GL(4)$ as $X^{\hat{m}\hat{n}} = (X^{0n} = Y^n,\ X^{mn}) \to X^{mn} = (X^{5\underline{n}} = x^{\underline{m}},\ X^{\underline{mn}} = y^{\underline{mn}})$ and $Y^m = (Y^5 = Y,\ Y^{\underline{m}})$. The 6-dimensional world-volume derivative is reduced into the 5-brane world-sheet derivatives as $\partial^{\hat{m}} = (\partial^0 = \partial_\tau,\ \partial^5 = 0,\ \partial^{\underline{m}} = \partial_\sigma{}^{\underline{m}})$. The SL(6) field strength for the $\mathcal{M}$5-brane has the following components

$$\begin{cases} \mathring{F}^{05\underline{m}} = \partial_\tau x^{\underline{m}} + \partial^{\underline{m}}Y\,, \\ \mathring{F}^{5\underline{mn}} = -\partial^{[\underline{m}}x^{\underline{n}]}\,, \\ \mathring{F}^{0\underline{mn}} = \partial_\tau y^{\underline{mn}} - \partial^{[\underline{m}}Y^{\underline{n}]}\,, \\ \mathring{F}^{\underline{mnl}} = \frac{1}{2}\partial^{[\underline{m}}y^{\underline{nl}]}\,, \end{cases} \tag{90}$$

where the auxiliary coordinates $y^{\underline{mn}}$ and $Y^m$ are preserved to begin with the SL(5) $A$-symmetric $\mathcal{M}$-theory Lagrangian [30].

The SL(6) vielbein for the $\mathcal{M}$5-brane with SL(5) $A$-symmetry is given by

$$E_{\hat{m}}{}^{\hat{a}} = \begin{pmatrix} E_0{}^{\hat{0}} & E_0{}^a \\ E_m{}^{\hat{0}} & E_m{}^a \end{pmatrix} = \begin{pmatrix} \dfrac{1}{g} & 0 \\ -\dfrac{s_m}{g} & g^{1/5}E_m{}^a \end{pmatrix}. \tag{91}$$

It is stressed that the world-volume vielbein fields $g$, $s_m$ and the spacetime vielbein $E_m{}^a$ cannot be in block diagonal form unlike $\mathcal{T}$-string case (75). The selfdual and anti-selfdual currents in curved background given by (32) based on (31) are the following combination of the SL(6) field strengths in (90) with (38) as

$$F_{\text{SD}/\overline{\text{SD}}}{}^{a_1 a_2} = g\left(F^{\hat{0}a_1 a_2} \pm \frac{1}{3!}\epsilon^{\hat{0}a_1 a_2}{}_{a_3 a_4 a_5}F^{a_3 a_4 a_5}\right). \tag{92}$$

The GL(4) covariant selfdual and anti-selfdual currents in flat space are derived from the ones of SL(5) (31) given in [1] as

$$\begin{cases} \mathring{F}_{\text{SD}/\overline{\text{SD}}}{}^{\underline{m}} &= F_\tau{}^{\underline{m}} - \frac{1}{2}\epsilon^{\underline{mn_1n_2n_3}}s_{\underline{n_1}}F_{\sigma;\underline{n_2n_3}} \pm g\hat{\eta}^{\underline{mn}}F_{\sigma;\underline{n}} \\ &= \varepsilon_\tau x^{\underline{m}} \pm g\hat{\eta}^{\underline{mn}}(\varepsilon_\sigma y)_{\underline{n}}\,, \\ \mathring{F}_{\text{SD}/\overline{\text{SD}}}{}^{\underline{m_1m_2}} &= F_\tau{}^{\underline{m_1m_2}} + \epsilon^{\underline{m_1}\cdots\underline{m_4}}s_{\underline{m_3}}F_{\sigma;\underline{m_4}} - \frac{1}{2}\epsilon^{\underline{m_1}\cdots\underline{m_4}}s_5 F_{\sigma;\underline{m_3m_4}} \pm g\hat{\eta}^{\underline{m_1n_1}}\hat{\eta}^{\underline{m_2n_2}}F_{\sigma;\underline{n_1n_2}} \\ &= \varepsilon_\tau y^{\underline{m_1m_2}} \pm g\hat{\eta}^{\underline{m_1n_1}}\hat{\eta}^{\underline{m_2n_2}}(\varepsilon_\sigma x)_{\underline{n_1n_2}}\,, \end{cases} \tag{93}$$

where $\hat{\eta}^{mn}$ becomes $G^{mn}$ in a curved background. The brane world-volume derivatives are

given as a generalization of the world-sheet zweibein dependence in (64) as

$$
\begin{cases}
\varepsilon_\tau x^{\underline{m}} & \equiv F_\tau{}^{\underline{m}} - \frac{1}{2}\epsilon^{\underline{m}n_1 n_2 n_3} s_{\underline{n_1}} F_{\sigma;\underline{n_2 n_3}} \\
& = \dot{x}^{\underline{m}} + \partial^{\underline{m}} Y + s_{\underline{n}} \partial^{[\underline{m}} x^{\underline{n}]}, \\
\varepsilon_\tau y^{\underline{m_1 m_2}} & \equiv F_\tau{}^{\underline{m_1 m_2}} + \epsilon^{\underline{m_1}\cdots\underline{m_4}} s_{\underline{m_3}} F_{\sigma;\underline{m_4}} \\
& = \dot{y}^{\underline{m_1 m_2}} - \partial^{[\underline{m_1}} Y^{\underline{m_2}]} + \frac{1}{2} s_{\underline{m_3}} \partial^{[\underline{m_1}} y^{\underline{m_2 m_3}]} + s_5 \partial^{[\underline{m_1}} x^{\underline{m_2}]},
\end{cases}
$$

$$
\begin{cases}
(\varepsilon_\sigma y)_{\underline{m}} & \equiv F_{\sigma;\underline{m}} \\
& = \frac{1}{2} \epsilon_{\underline{m}n_1 n_2 n_3} \partial^{\underline{n_1}} y^{\underline{n_2 n_3}}, \\
(\varepsilon_\sigma x)_{\underline{m_1 m_2}} & \equiv F_{\sigma;\underline{m_1 m_2}} \\
& = -\epsilon_{\underline{m_1}\cdots\underline{m_4}} \partial^{\underline{m_3}} x^{\underline{m_4}}.
\end{cases}
\tag{94}
$$

The 11-dimensional supergravity background includes the gravitational metric $g_{mn}$ and the three form gauge field $C_{mnl}$. We focus on the 4-dimensional subspace of the 11-dimensional space, where the background fields are $g_{\underline{mn}}$ and $C_{\underline{mnl}}$ whose number of degrees of freedom is $10+4 = 14$. The dimension of the coset $\mathrm{SL}(5)/\mathrm{SO}(5)$ is also $24-10 = 14$. The vector vielbein $E_m{}^a \in \mathrm{SL}(5)/\mathrm{SO}(5)$ with GL(4) indices where $m = (5,\underline{m})$ and $\underline{m} = 1,\cdots,4$ is given by [65]

$$
E_m{}^a = \begin{pmatrix} E_5{}^5 & E_5{}^{\underline{a}} \\ E_{\underline{m}}{}^5 & E_{\underline{m}}{}^{\underline{a}} \end{pmatrix} = \begin{pmatrix} \mathbf{e}^{3/5} & \mathbf{e}^{-2/5}\tilde{C}^{\underline{n}} e_{\underline{n}}{}^{\underline{a}} \\ 0 & \mathbf{e}^{-2/5} e_{\underline{m}}{}^{\underline{a}} \end{pmatrix},
$$

$$
\tilde{C}^{\bar{m}} = \frac{1}{3!} \epsilon^{\underline{m}\underline{m_1}\underline{m_2}\underline{m_3}} C_{\underline{m_1}\underline{m_2}\underline{m_3}},
\tag{95}
$$

$$
\mathbf{e} = \det e_{\underline{m}}{}^{\underline{a}},
$$

with $\det E_m{}^a = 1 = \epsilon^{m_1\cdots m_5} E_{m_1}{}^{a_1} E_{m_2}{}^{a_2} E_{m_3}{}^{a_3} E_{m_4}{}^{a_4} E_{m_5}{}^{a_5} = \epsilon^{a_1\cdots a_5}$. The tensor vielbein is the product of the vector vielbein (95) as

$$
E_{m_1 m_2}{}^{a_1 a_2} = E_{[m_1}{}^{a_1} E_{m_2]}{}^{a_2}
$$

$$
= \begin{pmatrix} E_{5\underline{m}}{}^{5\underline{a}} & E_{5\underline{m}}{}^{a_1 a_2} \\ E_{\underline{m_1}\underline{m_2}}{}^{5\underline{a}} & E_{\underline{m_1}\underline{m_2}}{}^{a_1 a_2} \end{pmatrix} = \begin{pmatrix} \mathbf{e}^{1/5} e_{\underline{m}}{}^{\underline{a}} & -\mathbf{e}^{-4/5}\tilde{C}^{\underline{n}} e_{\underline{m}}{}^{[\underline{a_1}} e_{\underline{n}}{}^{\underline{a_2}]} \\ 0 & \mathbf{e}^{-4/5} e_{\underline{m_1}}{}^{[\underline{a_1}} e_{\underline{m_2}}{}^{\underline{a_2}]} \end{pmatrix}.
\tag{96}
$$

The background gauge field in tensor index is now

$$
G_{MN} = G_{m_1 m_2; n_1 n_2} = \frac{1}{2^2} E_{m_1 m_2}{}^{a_1 a_2} \hat{\eta}_{a_1[b_1} \hat{\eta}_{b_2]a_2} E_{n_1 n_2}{}^{b_1 b_2}
$$

$$
= \begin{pmatrix} G_{\underline{mn}} & G_{\underline{m};\underline{n_1 n_2}} \\ G_{\underline{m_1 m_2};\underline{n}} & G_{\underline{m_1 m_2};\underline{n_1 n_2}} \end{pmatrix}
$$

$$
= \mathbf{e}^{-8/5} \begin{pmatrix} \mathbf{e}^2 g_{\underline{mn}} - \tilde{C}^{\underline{p}} g_{\underline{m}[\underline{p}} g_{\underline{n}]\underline{q}} \tilde{C}^{\underline{q}} & \tilde{C}^{\underline{l}} g_{\underline{l}[\underline{n_1}} g_{\underline{n_2}]\underline{m}} \\ g_{\underline{p}[\underline{m_1}} g_{\underline{m_2}]\underline{n}} \tilde{C}^{\underline{p}} & g_{\underline{m_1}[\underline{n_1}} g_{\underline{n_2}]\underline{m_2}} \end{pmatrix}
$$

$$
= \mathbf{e}^{2/5} \begin{pmatrix} g_{\underline{mn}} & 0 \\ 0 & 0 \end{pmatrix} + \mathbf{e}^{-8/5} \begin{pmatrix} -\tilde{C}^{\underline{p}} g_{\underline{m}[\underline{p}} g_{\underline{n}]\underline{q}} \tilde{C}^{\underline{q}} & \tilde{C}^{\underline{l}} g_{\underline{l}[\underline{n_1}} g_{\underline{n_2}]\underline{m}} \\ g_{\underline{p}[\underline{m_1}} g_{\underline{m_2}]\underline{n}} \tilde{C}^{\underline{p}} & g_{\underline{m_1}[\underline{n_1}} g_{\underline{n_2}]\underline{m_2}} \end{pmatrix}.
\tag{97}
$$

Inverse of these background gauge fields are given by [65]

$$
E_a{}^m = \begin{pmatrix} E_5{}^5 & E_5{}^{\underline{m}} \\ E_{\underline{a}}{}^5 & E_{\underline{a}}{}^{\underline{m}} \end{pmatrix} = \begin{pmatrix} \mathbf{e}^{-3/5} & \mathbf{e}^{-3/5}\tilde{C}^{\underline{m}} \\ 0 & \mathbf{e}^{2/5}e_{\underline{a}}{}^{\underline{m}} \end{pmatrix}, \tag{98}
$$

$$
E_{a_1a_2}{}^{m_1m_2} = E_{[a_1}{}^{m_1}E_{a_2]}{}^{m_2}
$$
$$
= \begin{pmatrix} E_{5\underline{a}}{}^{5\underline{m}} & E_{5\underline{a}}{}^{\underline{m_1m_2}} \\ E_{\underline{a_1a_2}}{}^{5\underline{m}} & E_{\underline{a_1a_2}}{}^{\underline{m_1m_2}} \end{pmatrix} = \begin{pmatrix} e_{\underline{a}}{}^{\underline{m}} & -\tilde{C}^{[\underline{m_1}}e_{\underline{a}}{}^{\underline{m_2}]} \\ 0 & \mathbf{e}\,e_{\underline{a_1}}{}^{[\underline{m_1}}e_{\underline{a_2}}{}^{\underline{m_2}]} \end{pmatrix}, \tag{99}
$$

$$
G^{MN} = G^{m_1m_2;n_1n_2} = \frac{1}{2^2}E_{a_1a_2}{}^{m_1m_2}\hat{\eta}^{a_1[b_1}\hat{\eta}^{b_2]a_2}E_{b_1b_2}{}^{n_1n_2}
$$
$$
= \begin{pmatrix} G^{\underline{mn}} & G^{\underline{m};\underline{n_1n_2}} \\ G^{\underline{m_1m_2};\underline{n}} & G^{\underline{m_1m_2};\underline{n_1n_2}} \end{pmatrix}
$$
$$
= \mathbf{e}^{2/5}\begin{pmatrix} g^{\underline{mn}} & g^{\underline{m}[\underline{n_1}}\tilde{C}^{\underline{n_2}]} \\ -\tilde{C}^{[\underline{m_1}}g^{\underline{m_2}]\underline{n}} & \mathbf{e}^2 g^{\underline{m_1}[\underline{n_1}}g^{\underline{n_2}]\underline{m_2}} + \tilde{C}^{[\underline{m_1}}g^{\underline{m_2}][\underline{n_1}}\tilde{C}^{\underline{n_2}]} \end{pmatrix}
$$
$$
= \mathbf{e}^{8/5}\begin{pmatrix} 0 & 0 \\ 0 & g^{\underline{m_1}[\underline{n_1}}g^{\underline{n_2}]\underline{m_2}} \end{pmatrix} + \mathbf{e}^{-2/5}\begin{pmatrix} g^{\underline{mn}} & g^{\underline{m}[\underline{n_1}}\tilde{C}^{\underline{n_2}]} \\ -\tilde{C}^{[\underline{m_1}}g^{\underline{m_2}]\underline{n}} & \tilde{C}^{[\underline{m_1}}g^{\underline{m_2}][\underline{n_1}}\tilde{C}^{\underline{n_2}]} \end{pmatrix}. \tag{100}
$$

The $\mathcal{M}5$-brane Lagrangian is given by the SL(5) covariant Lagrangian (33) with replacing GL(4) indices as

$$
L = \frac{1}{2}\phi\left(F_{\text{SD}}{}^{\underline{a}}F_{\overline{\text{SD}}\underline{a}} + \frac{1}{2}F_{\text{SD}}{}^{\underline{ab}}F_{\overline{\text{SD}}\underline{ab}}\right) + \frac{1}{2}\bar{\phi}\left((F_{\overline{\text{SD}}}{}^{\underline{a}})^2 + \frac{1}{2}(F_{\overline{\text{SD}}}{}^{\underline{ab}})^2\right)
$$
$$
+ \frac{1}{2}\lambda F_{\text{SD}}{}^{\underline{c}}F_{\overline{\text{SD}}\underline{c}} + \lambda_{\underline{a}}F_{\overline{\text{SD}}}{}^{\underline{ac}}F_{\overline{\text{SD}}\underline{c}} + \frac{1}{2}\lambda_{\underline{ab}}\left(F_{\overline{\text{SD}}}{}^{\underline{a}}F_{\overline{\text{SD}}}{}^{\underline{b}} + F_{\overline{\text{SD}}}{}^{\underline{ac}}F_{\overline{\text{SD}}}{}^{\underline{b}}{}_{\underline{c}}\right) \tag{101}
$$
$$
- \epsilon_{\underline{a_1}\cdots\underline{a_4}}\left(\frac{1}{8}\lambda^5 F_{\overline{\text{SD}}}{}^{\underline{a_1a_2}}F_{\overline{\text{SD}}}{}^{\underline{a_3a_4}} - \frac{1}{2}\lambda^{\underline{a_1}}F_{\overline{\text{SD}}}{}^{\underline{a_2a_3}}F_{\overline{\text{SD}}}{}^{\underline{a_4}}\right).
$$

The Lagrangian in terms of the curved currents $F_{\text{SD}/\overline{\text{SD}}}^{ab}$ is simpler than the one in terms of the flat currents $\mathring{F}_{\text{SD}/\overline{\text{SD}}}{}^{mn}$. The concrete expression of the Lagrangian of the $\mathcal{M}5$-brane in a curved background (101) is given as follows. We begin by the SL(5) covariant Lagrangian (33) in the gauge $\phi = \frac{1}{2g}$ and $\bar{\phi} = 0 = \lambda$'s

$$
\frac{1}{2g}F_{\text{SD}}{}^A\hat{\eta}_{AB}F_{\overline{\text{SD}}}{}^B = \frac{1}{8g}\mathring{F}_{\text{SD}}{}^{m_1m_2}G_{m_1m_2;n_1n_2}\mathring{F}_{\overline{\text{SD}}}{}^{n_1n_2}
$$
$$
= \frac{1}{8g}\varepsilon_\tau X^{m_1m_2}G_{m_1m_2;n_1n_2}\varepsilon_\tau X^{n_1n_2} - \frac{g}{8}F_{\sigma m_1m_2}G^{m_1m_2;n_1n_2}F_{\sigma n_1n_2}
$$
$$
= \frac{1}{2g}(\varepsilon_\tau x^{\underline{m}}\varepsilon_\tau y^{\underline{m_1m_2}})\Bigg[\mathbf{e}^{2/5}\begin{pmatrix} g_{\underline{mn}} & 0 \\ 0 & 0 \end{pmatrix}
$$
$$
+ \mathbf{e}^{-8/5}\begin{pmatrix} -\tilde{C}^{\underline{p}}g_{\underline{m}[\underline{p}}g_{\underline{n}]\underline{q}}\tilde{C}^{\underline{q}} & \tilde{C}^{\underline{l}}g_{\underline{l}[\underline{n_1}}g_{\underline{n_2}]\underline{m}} \\ g_{\underline{p}[\underline{m_1}}g_{\underline{m_2}]\underline{n}}\tilde{C}^{\underline{p}} & g_{\underline{m_1}[\underline{n_1}}g_{\underline{n_2}]\underline{m_2}} \end{pmatrix}\Bigg]\begin{pmatrix} \varepsilon_\tau x^{\underline{n}} \\ \varepsilon_\tau y^{\underline{n_1n_2}} \end{pmatrix} \tag{102}
$$
$$
- \frac{g}{2}\left((\varepsilon_\sigma y)_{\underline{m}}(\varepsilon_\sigma x)_{\underline{m_1m_2}}\right)\Bigg[\mathbf{e}^{8/5}\begin{pmatrix} 0 & 0 \\ 0 & g^{\underline{m_1}[\underline{n_1}}g^{\underline{n_2}]\underline{m_2}} \end{pmatrix}
$$
$$
+ \mathbf{e}^{-2/5}\begin{pmatrix} g^{\underline{mn}} & g^{\underline{m}[\underline{n_1}}\tilde{C}^{\underline{n_2}]} \\ -\tilde{C}^{[\underline{m_1}}g^{\underline{m_2}]\underline{n}} & \tilde{C}^{[\underline{m_1}}g^{\underline{m_2}][\underline{n_1}}\tilde{C}^{\underline{n_2}]} \end{pmatrix}\Bigg]\begin{pmatrix} (\varepsilon_\sigma y)_{\underline{n}} \\ (\varepsilon_\sigma x)_{\underline{n_1n_2}} \end{pmatrix}
$$
$$
= L_0 + L_y,
$$

with

$$L_0 = \frac{\mathbf{e}^{2/5}}{2g}\varepsilon_\tau x^{\underline{m}}g_{\underline{mn}}\varepsilon_\tau x^{\underline{n}} - \frac{g\mathbf{e}^{8/5}}{8}(\varepsilon_\sigma x)_{\underline{m_1 m_2}}g^{\underline{m_1}[\underline{n_1}}g^{\underline{n_2}]\underline{m_2}}(\varepsilon_\sigma x)_{\underline{n_1 n_2}},$$

$$L_y = \frac{\mathbf{e}^{-8/5}}{8g}\left(\varepsilon_\tau y^{\underline{m_1 m_2}} - \varepsilon_\tau x^{[\underline{m_1}}\tilde{C}^{\underline{m_2}]}\right)g_{\underline{m_1}[\underline{n_1}}g_{\underline{n_2}]\underline{m_2}}\left(\varepsilon_\tau y^{\underline{n_1 n_2}} + \tilde{C}^{[\underline{n_1}}\varepsilon_\tau x^{\underline{n_2}]}\right) \tag{103}$$

$$- \frac{g\mathbf{e}^{-2/5}}{2}\left((\varepsilon_\sigma y)_{\underline{m}} + (\varepsilon_\sigma x)_{\underline{ml}}\tilde{C}^{\underline{l}}\right)g^{\underline{mn}}\left((\varepsilon_\sigma y)_{\underline{n}} - \tilde{C}^{\underline{l}}(\varepsilon_\sigma x)_{\underline{ln}}\right).$$

In the gauge $g = \mathbf{e}^{-3/5}$ Lagrangians take simple form as

$$L_0 = -\frac{\mathbf{e}}{2}\left[F^{05\underline{a}}\hat{\eta}_{\underline{ab}}F^{05\underline{b}} - \frac{1}{4}F^{5\underline{a_1 a_2}}\hat{\eta}_{\underline{a_1}[\underline{b_1}}\hat{\eta}_{\underline{b_2}]\underline{a_2}}F^{5\underline{b_1 b_2}}\right],$$

$$L_y = -\frac{1}{2\mathbf{e}}\left[\frac{1}{2}F^{0\underline{a_1 a_2}}\hat{\eta}_{\underline{a_1 b_1}}\hat{\eta}_{\underline{b_2 a_2}}F^{0\underline{b_1 b_2}} - \frac{1}{6}F^{\underline{a_1 a_2 a_3}}\hat{\eta}_{\underline{a_1 b_1}}\hat{\eta}_{\underline{a_2 b_2}}\hat{\eta}_{\underline{a_3 b_3}}F^{\underline{b_1 b_2 b_3}}\right], \tag{104}$$

with $\tilde{m} = (0, \underline{m})$.

The SL(5) U-duality symmetry of the Lagrangian (104) is broken to GL(4) symmetry by the dimensional reduction similarly to (2). Then the kinetic term of the new perturbative Lagrangian for a $\mathcal{M}$5-brane in the 4-dimensions is given by

$$L_0 = \frac{\mathbf{e}}{2}\left[(\dot{x}^{\underline{m}} + \partial^{\underline{m}}Y + s_{\underline{l}}\partial^{[\underline{m}}x^{\underline{l}]})g_{\underline{mn}}(\dot{x}^{\underline{n}} + \partial^{\underline{n}}Y + s_{\underline{k}}\partial^{[\underline{n}}x^{\underline{k}]})\right.$$

$$\left. - \frac{1}{4}\partial^{[\underline{m_1}}x^{\underline{m_2}]}g_{\underline{m_1}[\underline{n_1}}g_{\underline{n_2}]\underline{m_2}}\partial^{[\underline{n_1}}x^{\underline{n_2}]}\right]. \tag{105}$$

The total derivative terms to obtain the Wess-Zumino term for the $\mathcal{M}$5-brane are given analogously to the string case (70) with the gauge $\partial^{\underline{m}}s_5 = 0$ as

$$\varepsilon_\tau x^{\underline{m}}(\varepsilon_\sigma y)_{\underline{m}} - \frac{1}{2}(\varepsilon_\sigma x)_{\underline{m_1 m_2}}\varepsilon_\tau y^{\underline{m_1 m_2}}$$

$$= \frac{1}{2}\epsilon_{\underline{m_1 \cdots m_4}}\left\{\partial_\tau(x^{\underline{m_1}}\partial^{\underline{m_2}}y^{\underline{m_3 m_4}})\right. \tag{106}$$

$$\left. + \partial^{\underline{m_1}}(x^{\underline{m_2}}\dot{y}^{\underline{m_3 m_4}} + Y\partial^{\underline{m_2}}y^{\underline{m_3 m_4}} - 2x^{\underline{m_2}}\partial^{\underline{m_3}}Y^{\underline{m_4}} - 2s_5 x^{\underline{m_2}}\partial^{\underline{m_3}}x^{\underline{m_4}})\right\},$$

where the $s_{\underline{n}}$ dependent terms are cancelled out because of the totally antisymmetricity of 5 indices

$$\epsilon_{\underline{m_1 \cdots m_4}}s_{\underline{n}}\left((\partial^{[\underline{m_1}}x^{\underline{n}]})\partial^{\underline{m_2}}y^{\underline{m_3 m_4}} + \frac{1}{2}(\partial^{\underline{m_4}}x^{\underline{m_1}})\partial^{[\underline{n}}y^{\underline{m_2 m_3}]}\right) = \epsilon_{\underline{m_1 \cdots m_4}}s_{\underline{n}}\frac{1}{4!}\partial^{[\underline{m_1}}x^{\underline{n}}\partial^{\underline{m_2}}y^{\underline{m_3 m_4}]} = 0. \tag{107}$$

Adding the total derivative term (106) to the $\mathcal{M}$5-brane Lagrangian (101) in gauge $\phi = \frac{1}{2g}$, $g = 2\mathbf{e}^{-3/5}$, $\bar{\phi} = 0 = \lambda$'s the Lagrangian for the $\mathcal{M}$5-brane becomes

$$\frac{1}{2g}F_{\mathrm{SD}}{}^A\hat{\eta}_{AB}F_{\mathrm{SD}}{}^B - \varepsilon_\tau x^{\underline{m}}(\varepsilon_\sigma y)_{\underline{m}} + \frac{1}{2}(\varepsilon_\sigma x)_{\underline{m_1 m_2}}\varepsilon_\tau y^{\underline{m_1 m_2}} = L_0 + L_y + L_{\mathrm{WZ}},$$

$$L_0 = \frac{\mathbf{e}}{4}\left[\varepsilon_\tau x^{\underline{m}}g_{\underline{mn}}\varepsilon_\tau x^{\underline{n}} - \frac{1}{4}(\varepsilon_\sigma x)_{\underline{m_1 m_2}}g^{\underline{m_1}[\underline{n_1}}g^{\underline{n_2}]\underline{m_2}}(\varepsilon_\sigma x)_{\underline{n_1 n_2}}\right],$$

$$L_y = \frac{1}{4\mathbf{e}}\left[\frac{1}{4}\left(\varepsilon_\tau y^{\underline{m_1 m_2}} - \varepsilon_\tau x^{[\underline{m_1}}\tilde{C}^{\underline{m_2}]} + \mathbf{e}(\varepsilon_\sigma x)_{\underline{l_1 l_2}}g^{\underline{l_1 m_1}}g^{\underline{l_2 m_2}}\right)g_{\underline{m_1}[\underline{n_1}}g_{\underline{n_2}]\underline{m_2}}\right.$$

$$\times \left(\varepsilon_\tau y^{\underline{n_1 n_2}} + \tilde{C}^{[\underline{n_1}}\varepsilon_\tau x^{\underline{n_2}]} + \mathbf{e}g^{\underline{n_1 k_1}}g^{\underline{n_2 k_2}}(\varepsilon_\sigma x)_{\underline{k_1 k_2}}\right)$$

$$\left. - \left((\varepsilon_\sigma y)_{\underline{m}} + (\varepsilon_\sigma x)_{\underline{ml}}\tilde{C}^{\underline{l}} + \mathbf{e}\varepsilon_\tau x^{\underline{l}}g_{\underline{lm}}\right)g^{\underline{mn}}\left((\varepsilon_\sigma y)_{\underline{n}} - \tilde{C}^{\underline{k}}(\varepsilon_\sigma x)_{\underline{kn}} + \mathbf{e}g_{\underline{nk}}\varepsilon_\tau x^{\underline{k}}\right)\right],$$

$$L_{\mathrm{WZ}} = \varepsilon_\tau x^{\underline{m_1}}\tilde{C}^{\underline{m_2}}(\varepsilon_\sigma x)_{\underline{m_1 m_2}}. \tag{108}$$

Dimensional reduction $L_y \to 0$ gives the $\mathcal{M}5$-brane Lagrangian with the Wass-Zumino term. The obtained new $\mathcal{M}5$-brane Lagrangian in the supergravity background (42) is

$$
\begin{aligned}
L_{\mathcal{M}5} &= L_0 + L_{\mathrm{WZ}}, \\
L_0 &= \frac{\mathsf{e}}{2}\bigg[ (\dot{x}^{\underline{m}} + \partial^{\underline{m}} Y + s_{\underline{l}} \partial^{[\underline{m}} x^{\underline{l}]}) g_{\underline{mn}} (\dot{x}^{\underline{n}} + \partial^{\underline{n}} Y + s_{\underline{k}} \partial^{[\underline{n}} x^{\underline{k}]}) \\
&\qquad - \frac{1}{4} \partial^{[\underline{m}_1} x^{\underline{m}_2]} g_{\underline{m}_1[\underline{n}_1} g_{\underline{n}_2]\underline{m}_2} \partial^{[\underline{n}_1} x^{\underline{n}_2]} \bigg], \\
L_{\mathrm{WZ}} &= (\dot{x}^{\underline{m}_1} + \partial^{\underline{m}_1} Y) C_{\underline{m}_1 \underline{m}_2 \underline{m}_3} \partial^{\underline{m}_2} x^{\underline{m}_3} + \frac{1}{6} (\partial^{\underline{m}_1} x^{\underline{m}_2})(\partial^{\underline{m}_3} x^{\underline{m}_4}) s_{[\underline{m}_1} C_{\underline{m}_2 \underline{m}_3 \underline{m}_4]}.
\end{aligned}
\tag{109}
$$

### 6.2 Non-perturbative M2-brane from $\mathcal{M}5$-brane

A non-perturbative membrane action in the 11-dimensional supergravity theory is given by [55]

$$
I = \int d^3\sigma \, L, \quad L = L_0 + L_{WZ},
$$

$$
\begin{cases}
L_0 = -T\sqrt{-\det \partial_\mu x^m \partial_\nu x^n g_{mn}}, \\
L_{\mathrm{WZ}} = \frac{T}{3!} \epsilon^{\mu\nu\rho} \partial_\mu x^{m_1} \partial_\nu x^{m_2} \partial_\rho x^{m_3} C_{m_1 m_2 m_3},
\end{cases}
\tag{110}
$$

with the spacetime index $m = 0, 1, \cdots, 10$ and the world-volume index $\mu = 0, 1, 2$. The canonical coordinates are $x^m$ and $p_m$, and the spacial world-volume coordinate derivative is $\partial_i$ with $i = 1, 2$. The Hamiltonian is given by [65] where $p_m = \partial L / \partial \dot{x}^m$

$$
\begin{aligned}
H &= p_m \dot{x}^m - L \\
&= \lambda_0 \mathcal{H}_\tau + \lambda^i \mathcal{H}_i, \\
&\begin{cases}
\mathcal{H}_\tau = \frac{1}{2} \triangleright_a \eta^{ab} \triangleright_b + \frac{1}{8} \triangleright^{a_1 a_2} \eta_{a_1[b_1} \eta_{b_2]a_2} \triangleright^{b_1 b_2}, \\
\mathcal{H}_i = \partial_i x^m p_m.
\end{cases}
\end{aligned}
\tag{111}
$$

Here $\triangleright_A = (\triangleright_a, \triangleright^{ab})$ is related to $\triangleright_M = (\triangleright_m = p_m, \triangleright^{mn} = \epsilon^{ij} \partial_i x^{m_1} \partial_j x^{m_2})$ as $\triangleright_A = E_A{}^M \triangleright_M$ for the background gauge field $E_A{}^M$. $E_A{}^M$ includes $g_{mn}$ and $C_{mnl}$. The Virasoro constraint $\mathcal{S}^m = 0$ in (19) is related to the constraint $\mathcal{H}_i = 0$ in (111) which generates $\sigma$-diffeomorphism by multiplying the world-volume embedding operator in (114) as $\mathcal{S}^m = \mathcal{H}_i \epsilon^{ij} \partial_j x^m$.

We focus on the 4-dimensional subspace where the supergravity background is a representation of the SL(5) U-duality symmetry, $E_A{}^M \in \mathrm{SL}(5)/\mathrm{SO}(5)$. The currents $\triangleright_{\underline{m}}$ and $\triangleright_{\underline{mn}}$ are 4 and 6 components of SL(4) with $\underline{m} = 1, \cdots, 4$, which are unified into a SL(5) tensor $\triangleright_{mn} = (\triangleright_{\underline{m}}, \triangleright_{\underline{mn}})$ with $m = 1, \cdots, 5$. The currents for a M2-brane in 4-dimensional space (111) obtained from the membrane Lagrangian (110) are written as

$$
\begin{cases}
\triangleright_{\underline{m}} = p_{\underline{m}}, \\
\triangleright_{\underline{m}_1 \underline{m}_2} = \frac{1}{2} \epsilon_{\underline{m}_1 \cdots \underline{m}_4} \epsilon^{ij} \partial_i x^{\underline{m}_3} \partial_j x^{\underline{m}_4}.
\end{cases}
\tag{112}
$$

Commutators of (112) are given as

$$
\begin{cases}
\left[ \triangleright_{\underline{m}}(\sigma), \triangleright_{\underline{n}}(\sigma') \right] = 0, \\
\left[ \triangleright_{\underline{m}_1}(\sigma), \triangleright_{\underline{m}_2 \underline{m}_3}(\sigma') \right] = 2i \epsilon_{\underline{m}_1 \cdots \underline{m}_4} \epsilon^{ij} \partial_j x^{\underline{m}_4} \partial_i \delta^{(2)}(\sigma - \sigma'), \\
\left[ \triangleright_{\underline{m}_1 \underline{m}_2}(\sigma), \triangleright_{\underline{m}_3 \underline{m}_4}(\sigma') \right] = 0.
\end{cases}
\tag{113}
$$

The $p$-brane current algebras with the non-perturbative winding modes $dx^{m_1} \wedge \cdots \wedge dx^{m_p}$ are obtained similarly in [66].

Now let us compare the SL(5) current algebra of the non-perturbative M2-brane (113) with the one of the $\mathcal{M}5$-brane (18). The perturbative $\mathcal{M}5$-brane current algebra in (18) reduces into the non-perturbative M2-brane algebra in (113) by reducing the 5-dimensional world-volume of the $\mathcal{A}5$-brane into the 2-dimensional world-volume of the non-perturbative M2-brane as

$$\partial^{\underline{m}} = \epsilon^{ij} \partial_j x^{\underline{m}} \partial_i. \tag{114}$$

The operator $\partial_j x^{\underline{m}}$ is an embedding of the membrane world-volume to the 5-brane world-volume (where the 5-th brane coordinate is in the internal space). It has the constant form $\partial_j x^{\underline{m}} = \delta^{\underline{m}}_j$ in the static gauge for the ground state [1].

Now we plug the world-volume projection (114) into the $\mathcal{M}5$-brane Lagrangian (109). The first term in the $Y = 0$ gauge is given by

$$
\begin{aligned}
(\dot{x}^{\underline{m}} + s_{\underline{l}} \epsilon^{ij} \partial_j x^{\underline{m}} \partial_i x^{\underline{l}}) & g_{\underline{mn}} (\dot{x}^{\underline{n}} + s_{\underline{k}} \epsilon^{i'j'} \partial_{j'} x^{\underline{n}} \partial_{i'} x^{\underline{k}}) \\
&= (\dot{x}^{\underline{a}})^2 + 2 s_{\underline{b}} \dot{x}_{\underline{a}} \epsilon^{ij} \partial_j x^{\underline{a}} \partial_i x^{\underline{b}} + (s_{\underline{b}} \epsilon^{ij} \partial_j x^{\underline{a}} \partial_i x^{\underline{b}})^2 \\
&= h_{00} - 2\lambda^i h_{0i} + \lambda^i \lambda^j h_{ij},
\end{aligned}
\tag{115}
$$

with

$$h_{ij} = \partial_i x^{\underline{a}} \partial_j x_{\underline{a}} = \partial_i x^{\underline{m}} g_{\underline{mn}} \partial_j x^{\underline{n}}, \qquad \partial_i x^{\underline{a}} \equiv e_{\underline{m}}{}^{\underline{a}} \partial_i x^{\underline{m}}, \tag{116}$$

and the membrane vielbein $\lambda^i$ and the 5-brane vielbein $s_a$

$$\lambda^i = s_{\underline{a}} \epsilon^{ij} \partial_j x^{\underline{a}}, \qquad s_{\underline{a}} = e_{\underline{a}}{}^{\underline{m}} s_{\underline{m}}. \tag{117}$$

The second term is given by

$$\frac{1}{2} \left( \frac{1}{2} \epsilon^{\underline{mn}}{}_{\underline{l_1 l_2}} \epsilon^{ij} \partial_j x^{l_1} \partial_i x^{l_2} \right)^2 = -\frac{1}{2} (\epsilon^{ij} \partial_j x^{\underline{m}} \partial_i x^{\underline{n}})^2 = -\det h_{ij}, \tag{118}$$

where the following relation is used in the last equality of (118)

$$\det h_{ij} = \frac{1}{2} \epsilon^{ii'} h_{ij} h_{i'j'} \epsilon^{jj'} = \frac{1}{2} \epsilon^{ii'} \partial_i x^{\underline{a}} \partial_j x_{\underline{a}} \partial_{i'} x^{\underline{b}} \partial_{j'} x_{\underline{b}} \epsilon^{jj'} = \frac{1}{2} (\epsilon^{ij} \partial_j x^{\underline{a}} \partial_i x^{\underline{b}})^2. \tag{119}$$

We choose the following gauge of the membrane world-volume metric

$$\phi = -\frac{\sqrt{-h} h^{00}}{2}, \qquad h = \det h_{\mu\nu}, \qquad \lambda^i = -\frac{h^{0i}}{h^{00}}, \qquad g^2 = \frac{-1}{h(h^{00})^2}. \tag{120}$$

Using with the relation

$$\det h_{ij} = h \, h^{00}, \tag{121}$$

the kinetic term $L_0$ in (109) becomes

$$
\begin{aligned}
L_0 &= \frac{1}{2} \left\{ -\sqrt{-h} - \sqrt{-h} \left( h^{00} h_{00} + 2 h^{0i} h_{0i} + \frac{h^{0i} h^{0j}}{(h^{00})^2} h_{ij} \right) \right\} \\
&= -\sqrt{-h}.
\end{aligned}
\tag{122}
$$

This is nothing but the Nambu-Goto Lagrangian for a membrane. The Wess-Zumino term $L_{\mathrm{WZ}}$ is obtained by using the world-volume projection (114) into (109) as

$$L_{\mathrm{WZ}} = \dot{x}^{\underline{m_1}} \epsilon^{ij} \partial_j x^{\underline{m_2}} \partial_i x^{\underline{m_3}} C_{\underline{m_1 m_2 m_3}} = \frac{1}{3!} \epsilon^{\mu\nu\rho} \partial_\mu x^{\underline{m_1}} \partial_\nu x^{\underline{m_2}} \partial_\rho x^{\underline{m_3}} C_{\underline{m_1 m_2 m_3}}. \tag{123}$$

Together with the Nambu-Goto term (122) the non-perturbative M2-brane Lagrangian is obtained from the perturbative $\mathcal{A}5$-brane as

$$
\begin{aligned}
I_{\mathrm{M2}} &= \int d^3\sigma\; L\,, \quad L = L_0 + L_{\mathrm{WZ}}\,, \\
L_0 &= -\sqrt{-\det \partial_\mu x^{\underline{m}}\partial_\nu x^{\underline{n}} g_{\underline{mn}}}\,, \\
L_{\mathrm{WZ}} &= \frac{1}{3!}\epsilon^{\mu\nu\rho}\partial_\mu x^{\underline{m_1}}\partial_\nu x^{\underline{m_2}}\partial_\rho x^{\underline{m_3}} C_{\underline{m_1 m_2 m_3}}\,.
\end{aligned}
\tag{124}
$$

This is the expected M2-brane Lagrangian (110) where we set $T = 1$.

## 7 Discussion

In this paper we have shown how the conventional strings and membrane are obtained from $\mathcal{A}$-theory five-brane with the SL(5) U-duality symmetry.

The following topics are interesting for future problems.

1. From $\mathcal{A}5$-brane to D-branes: The $\mathcal{A}$-theory background vielbein field includes the R-R gauge fields which couple to D-branes. The Nambu-Goto Lagrangian will be obtained analogously to the non-perturbative M2-brane Lagrangian as in subsection 6.2 with special care of the $B$-field. The Wess-Zumino term will be obtained by adding total derivative term with the $B$-field cloud, in such a way that the gauge transformation rule of the R-R gauge field involves the $B$-field.

2. From $\mathcal{A}$-theory branes to the non-perturbative $\mathcal{M}5$- and NS5-branes: The superstring theories admit the NS5-brane solutions which couple to the $B$-field magnetically. M-theory features the $\mathcal{M}5$-brane whose U-duality symmetry is realized by the current algebra [67], while type IIB superstring theory contains both the NS5-brane and D5-brane related by S-duality. These 5-brane Lagrangians are expected to be derived from $\mathcal{A}5$-brane and all such 5-branes should be connected via duality transformations. It is interesting to clarify the structure of the 5-brane WEB including $\mathcal{A}5$-brane for Lagrangians analogous to the one for current algebras [68].

3. From open $\mathcal{A}$-theory branes to heterotic strings and type I string: The Lagrangians of open $\mathcal{A}$-theory branes [26], which involve the SO(32) and E8×E8 gauge groups, as well as other half-BPS branes, are of particular interest.

4. Quantization of $\mathcal{A}$- and $\mathcal{M}$-branes: The main motivation for constructing the perturbative $\mathcal{A}$-brane Lagrangian is to facilitate a simpler quantization procedure. Quantum effects in string theory, including winding modes of strings and branes, play a crucial role in understanding Planck-scale physics, such as the resolution of the early-universe singularity. Quantizing $\mathcal{A}$-theory may provide valuable insights into a unified description of string spectra and S-matrices [69–72].

5. Higher dimensional cases: $\mathcal{A}$-theories in dimensions D>3 possess U-duality symmetry $E_{D+1}$ [46,73–75]. In these cases, the spacetime and world-volume dimensions become so large that they necessitate a new interpretation of the unphysical components of spacetime and world-volume. The construction of $\mathcal{A}$-theory may offer a new perspective on the fundamental description of string theory.

## Acknowledgments

We are grateful to Olaf Hohm, Igor Bandos, Martin Roček and Yuqi Li for the fruitful discussions. M.H. would like to thank Yuho Sakatani for useful discussions. We also acknowledge the Simons Center for Geometry and Physics for its hospitality during "The Simons Summer Workshop in Mathematics and Physics 2023 and 2024" where this work has been developed.

**Funding information** W.S. is supported by NSF award PHY-2210533. M.H. is supported in part by Grant-in-Aid for Scientific Research (C), JSPS KAKENHI Grant Numbers JP22K03603 and JP20K03604.

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
