# Peer review of "Strings and membranes from A-theory five brane"

_SciPost Physics, doi:SciPost Phys. 19, 009 (2025)_

## Round 1 · Referee Report · Anonymous (Referee 1) · 2025-4-17

Strengths
1) The authors continue the investigation of their previous work (reference [1] in the paper), which offers a novel perspective on world-volume theories and duality symmetry in string and M-theory. Their goal here is to construct a world-volume theory that is (manifestly) duality invariant under U-duality, analogous to how Exceptional Field Theory manifests the U-duality of maximal supergravity or how F-theory makes the SL(2) symmetry of IIB supergravity manifest. They propose a new 5-brane theory in an auxiliary 10-dimensional target space, which they call the A5-brane.
2) A particularly interesting and, to my knowledge, novel observation is that the SL(5) U-duality symmetry can be enhanced to SL(6), such that the field content not only incorporates the target space geometry but also includes a world-volume geometry of a 5-brane.
3) The authors clearly demonstrate that their new A5-brane theory reproduces known world-volume theories (without manifest duality symmetry) — such as those of the string, M2-brane, and M5-brane — upon dimensional reduction of both the world-volume and target space.
4) The draft is, in general, well-written and understandable, with sufficient technical detail and a solid introduction/overview of the topic and scope of the paper.
5) The references provide a well-rounded representation of the relevant literature.
Weaknesses
1) Since the authors focus on the low-dimensional case (D=3 in their notation), their construction results in an M5-brane theory in 4-dimensional space, which seems unusual. This raises the concern that their findings might be an artifact of the low-dimensional setting, which is not clearly addressed in the paper.
2) The paper does not engage in sufficient detail with alternative or related approaches to the problem of U-duality covariance in world-volume theories.
3) The quality of the English and general editorial state of the manuscript is subpar. In an era where automated tools can assist with grammar and style checking, the presence of numerous typos and syntactic issues that hinder readability is unnecessary and should be corrected.
A few examples:
Page 6, last paragraph of section 1.2: "In section 6 we begin by the SL(6) covariant A5-brane Lagrangian to lead the new perturbative M5-brane Lagrangian." -> Suggested revision: "In section 6, we begin with the SL(6)-covariant A5-brane Lagrangian, which leads to a new perturbative M5-brane Lagrangian."
Page 38, point 4: "Quantizaion" --> "Quantization"
Page 38, point 5, last sentence: "Constructing A-theory may give a hint of new description of string." -> Suggested revision: "Constructing A-theory may provide a hint toward a new description of string theory."
4) As in the authors’ previous work (or collaborations among them), the paper uses a notation that diverges from the conventions used by much of the double field theory / exceptional field theory community. While their notation appears consistent and in some cases predates others in the literature, this deviation may hinder accessibility and readability.
Report
Requested changes
1) Clarify dimensional dependence and generalisation potential: The authors should more clearly explain the implications of working in low dimensions (D=3), particularly how they obtain an M5-brane theory in 4-dimensional space, which seems nonstandard. They should also discuss whether the enhancement from SL(5) to SL(6) is a feature unique to low-dimensional settings, or if similar enhancements are expected for higher dimensions. Relevant literature discussing enhanced dualities in low-dimensional brane settings should be cited and discussed, e.g.:
M. Duff and J. Lu, Duality rotations in membrane theory, Nucl. Phys. B347 (1990) 394–419
M.J. Duff, J.X. Lu, R. Percacci, C.N. Pope, H. Samtleben, and E. Sezgin, Membrane Duality Revisited, [arXiv:1509.02915]
2) Discuss relation to existing approaches: Although the literature is broadly cited, a more explicit discussion of how the authors’ approach compares with other attempts to formulate U-duality covariant world-volume theories is needed. For example, a discussion of the differences and connections with works such as [47] and [62] (as numbered in their paper) would significantly strengthen the context and relevance of the present work.
3) Thorough proofreading and language polishing: The manuscript should be carefully proofread to correct grammatical and typographical errors, which currently detract from the clarity and flow of the presentation.
Recommendation
Ask for minor revision

---

## Round 1 · Referee Report · Anonymous (Referee 2) · 2025-4-28

Strengths
1) The paper explains how the A-theory fivebrane action with manifest SL(6) symmetry, which the authors originally wrote down in reference [1], reduces to (the actions for) other known lower-dimensional branes expressed in duality invariant formulations, namely strings and two-branes. This is an important consistency check on this proposed A-theory formulation.
2) Calculations and results are given rather explicitly.
Weaknesses
1) Science-wise: I am puzzled at the derivation of the "non-perturbative" M2 brane action from the A-theory fivebrane action via the "perturbative" M5 action (6.20). (Here "perturbative" means to quadratic order in derivatives and the "non-perturbative" M2 brane action is the usual Nambu-Goto style one, which features arbitrary orders in derivatives via a determinant in the lagrangian.) This apparent tension should be addressed or at least commented on.
2) Text-wise: the writing as well as the language is below par overall. The other report lists a few examples of typos and grammar/syntax issues so I will not repeat those here. There are also instances where text is duplicative or near duplicative of other text. (One that jumped at me was in the last paragraph of sec 1.1 and the first one of sec 1.2.) The arguments of this work were mostly clear, at least to this reader. However the text itself served to distract from rather than clarify and enhance the logical development of the material and I would recommend to the authors that future manuscripts should get more editorial attention both to the language of individual paragraphs as well as to how the text coheres as a whole. I found myself thinking "this was written by committee" a lot.
3) A comparatively minor quibble is that there could have been more of an effort to connect to the notation and terminology used by the Exceptional Field Theory subtopic to which this manuscript ostensibly contributes (as noted also by the other referee). In particular the use of \vartriangleright (I think) for currents is a bit distracting because one would naturally associate that symbol with e.g. a group action.
Report
Recommendation
Ask for minor revision

---

## Round 2 · Author Response

Warnings issued while processing user-supplied markup:
- Inconsistency: plain/Markdown and reStructuredText syntaxes are mixed. Markdown will be used.
Add "#coerce:reST" or "#coerce:plain" as the first line of your text to force reStructuredText or no markup.
You may also contact the helpdesk if the formatting is incorrect and you are unable to edit your text.
We thank the referees and the editorial team for their constructive feedback. We have revised the manuscript accordingly and addressed all comments and suggestions. A detailed point-by-point reply to the referees is included as a separate document. We hope that the revised version meets the expectations for publication in SciPost.
======================================== Reply to the referees
We would like to sincerely thank the reviewer for carefully reading our manuscript and providing thoughtful and constructive comments. Please find below the points we have revised for your review.
================================================================================================= Report #1
Weaknesses 1) → changes 1) 2) → changes 2) 3) → typos and grammar errors were corrected.
Requested changes 1) Clarify dimensional dependence and generalization potential:
① Implications of working in low dimensions (D=3) ⇒Page 4, line 10 from the bottom: In the last paragraph of subsection 1.1, the following explanation is added.
In the case of D=3, the theory provides a nontrivial yet tractable example that includes various types of branes and permits explicit computations. The Virasoro algebra, when extended to incorporate brane degrees of freedom, involves Gauss law–type constraints that are intrinsic to the brane. These constraints facilitate the dimensional reduction of the brane world-volume. Such an extended Virasoro algebra serves as a useful prototype for generalization to higher dimensions. Furthermore, the construction of the Lagrangian in this framework constitutes a significant milestone toward extending the formulation to higher-dimensional cases.
② An M5-brane in 4-dimensional space ⇒Page 13, line 8: The following explanation is added.
This M5-brane extends over both the main space (i.e., the duality-covariant space) and the internal space. Four of its world-volume directions lie in the main space while the remaining directions lie in the internal space, specifically one world-volume direction in the Hamiltonian formalism, or two in the Lagrangian formalism. Considering the critical string action in the full spacetime structure is an interesting subject, although it lies beyond the scope of the present discussion. The relationship between the main space and the internal space is schematically illustrated in Figure 2 the ``slug diagram" (see page 27 of arXiv:1610.0162 or page 14 of \cite{Hatsuda:2023dwx}). In the case of D = 3 the main space coordinate is represented by a bispinor $X^{\alpha\beta}$, and the world-volume coordinate by an antisymmetric bispinor $\sigma^{[\alpha\beta]}$, with $\alpha = 1, \dots, 4$. The internal space coordinate is given by a bispinor $Y^{[\alpha'\beta']}$, where $\alpha' = 1, \dots, 8$. The total number of supersymmetries is 32, which corresponds to the product of the dimensions of the spinor indices $32 = 4 \times 8$.
It is noted that the assignment of the duality symmetric space in ${\cal A}$-theory differs from that in conventional formulations. In ${\cal A}$-theory, the duality-symmetric space is assigned to the main "spacetime" rather than the internal space, such that all tensor gauge fields are automatically incorporated into the coset parameter of $\mathrm{E}_{\mathrm{D+1}} / H$.
③ Enhancement from SL(5) to SL(6) for higher dimenisons ⇒Page 5, line 10 from the bottom: In the third paragraph of subsection 1.2, the following explanation is added.
In general, the symmetry of Lagrangian formulation is larger than that of the corresponding Hamiltonian formulation. In ${\cal A}$-theory, the U-duality symmetry in the Hamiltonian formulation, G-symmetry, is enhanced to a novel duality symmetry in the Lagrangian formulation, A-symmetry. This symmetry enhancement in higher-dimensional cases (D < 6) is summarized on page 6 of arXiv:1806.02423 and page 14 of arXiv:2307.04934. It was shown that the brane world-volume metric is also transformed conformally under the SL(5) duality transformation as well as the spacetime background fields in \cite{Duff and Lu (1990) and Duff et al. in 1509.02915}. This mixing between spacetime and world-volume is a manifestation of the extended SL(6) duality symmetry transformation.
④ Relevant literature discussing enhanced dualities in low-dimensional brane ⇒Page 5, line 11 from the bottom: In the third paragraph of subsection 1.2, these references are cited.
Duff and Lu \cite{Duff Lu 1990, Duff et al 1509.02915} showed that the membrane theory exhibits the SL(5) duality symmetry by the Gaillard-Zumino approach.
2) Discuss relation to existing DFT/EFT approaches: ⇒Page 4, line 18: At the second-to-last paragraph of subsection 1.1, the following explanation is added.
The generalized diffeomorphism in EFT is characterized by the "Y-tensor", which reflects the structure of the exceptional group.
This Y-tensor, $Y^{MN}{}_{PQ}$, is related to the group-invariant metric in ${\cal A}$-theory, denoted by $\eta^{MNm}$, through the relation
$Y^{MN}{}{PQ} = \eta^{MNm} \eta$ where $M$ is the spacetime index and $m$ is the world-volume index. These indices correspond to different representations of the exceptional group.
The origin of the Y-tensor lies in the Schwinger term of the current algebra, ${\dd_N(\sigma),\dd_L(\sigma')}\sim \eta_{NLm}\partial^m
\delta(\sigma-\sigma')$,
where the world-volume derivative $\partial^m$ is defined through the commutator with the Virasoro constraint, ${\cal S}^m\sim \dd_N\eta^{NLm}\dd_L$.
The section condition given by the Y-tensor, $Y^{MN}{}_{PQ} \partial_M \partial_N = 0$, is related to the zero-mode condition of the Virasoro constraint in ${\cal A}$-theory.
Specifically, the zero-mode component of the constraint ${\cal S}^m$ takes the form ${\cal S}^m|_{\rm 0\text{-}modes} = \eta^{MNm} \partial_M \partial_N = 0,$
establishing a direct connection between the section condition in EFT and the Virasoro structure of ${\cal A}$-theory.
================================================================================================= Report #2 Weakness 1) Derivation of "non-perturbative" M2-brane from "perturbative" M5-brane: ⇒Page 6, line 14: In the last paragraph of subsection 1.2, the following explanation is added.
The perturbative" M5-brane Lagrangian is formulated as a bilinear expression in terms of currents, while thenon-perturbative" M2-brane Lagrangian comprises the sum of the Nambu–Goto and Wess–Zumino terms, each exhibiting multilinear dependence on the spacetime coordinates.
The dimensional reduction from the M5-brane to the M2-brane is implemented via
the ``non-perturbative projection" $ \partial^m = \epsilon^{ij} \partial_j x^m \partial_i, $
in \bref{stwvMix} and the gauge fixing of the world-volume metric in \bref{gaugechoicewv}.
2) English language ⇒ We have revised the English to the best of our ability.
3) Notation ⇒We acknowledge that denoting the stringy covariant derivative with a tilted symbol $\tilde{\nabla}$ may be somewhat unconventional. We kindly ask for the reader’s understanding due to the limitation of available notation.

---

## Round 2 · List of Changes

List of changes
- Page 4, line 10 from the bottom: In the last paragraph of subsection 1.1, the following explanation is added.
In the case of D=3, the theory provides a nontrivial yet tractable example that includes various types of branes and permits explicit computations. The Virasoro algebra, when extended to incorporate brane degrees of freedom, involves Gauss law–type constraints that are intrinsic to the brane. These constraints facilitate the dimensional reduction of the brane world-volume. Such an extended Virasoro algebra serves as a useful prototype for generalization to higher dimensions. Furthermore, the construction of the Lagrangian in this framework constitutes a significant milestone toward extending the formulation to higher-dimensional cases.
- Page 4, line 18: At the second-to-last paragraph of subsection 1.1, the following explanation is added.
The generalized diffeomorphism in EFT is characterized by the "Y-tensor", which reflects the structure of the exceptional group.
This Y-tensor, $Y^{MN}{}_{PQ}$, is related to the group-invariant metric in ${\cal A}$-theory, denoted by $\eta^{MNm}$, through the relation
$Y^{MN}{}{PQ} = \eta^{MNm} \eta$ where $M$ is the spacetime index and $m$ is the world-volume index. These indices correspond to different representations of the exceptional group.
The origin of the Y-tensor lies in the Schwinger term of the current algebra, ${\dd_N(\sigma),\dd_L(\sigma')}\sim \eta_{NLm}\partial^m
\delta(\sigma-\sigma')$,
where the world-volume derivative $\partial^m$ is defined through the commutator with the Virasoro constraint, ${\cal S}^m\sim \dd_N\eta^{NLm}\dd_L$.
The section condition given by the Y-tensor, $Y^{MN}{}_{PQ} \partial_M \partial_N = 0$, is related to the zero-mode condition of the Virasoro constraint in ${\cal A}$-theory.
Specifically, the zero-mode component of the constraint ${\cal S}^m$ takes the form ${\cal S}^m|_{\rm 0\text{-}modes} = \eta^{MNm} \partial_M \partial_N = 0,$
establishing a direct connection between the section condition in EFT and the Virasoro structure of ${\cal A}$-theory.
- Page 5, line 10 from the bottom: In the third paragraph of subsection 1.2, the following explanation is added.
In general, the symmetry of Lagrangian formulation is larger than that of the corresponding Hamiltonian formulation. In ${\cal A}$-theory, the U-duality symmetry in the Hamiltonian formulation, G-symmetry, is enhanced to a novel duality symmetry in the Lagrangian formulation, A-symmetry. This symmetry enhancement in higher-dimensional cases (D < 6) is summarized on page 6 of arXiv:1806.02423 and page 14 of arXiv:2307.04934. It was shown that the brane world-volume metric is also transformed conformally under the SL(5) duality transformation as well as the spacetime background fields in \cite{Duff and Lu (1990) and Duff et al. in 1509.02915}. This mixing between spacetime and world-volume is a manifestation of the extended SL(6) duality symmetry transformation.
- Page 5, line 11 from the bottom: In the third paragraph of subsection 1.2, these references are cited.
Duff and Lu \cite{Duff Lu 1990, Duff et al 1509.02915} showed that the membrane theory exhibits the SL(5) duality symmetry by the Gaillard-Zumino approach.
- Page 6, line 14: In the last paragraph of subsection 1.2, the following explanation is added.
The perturbative" M5-brane Lagrangian is formulated as a bilinear expression in terms of currents, while thenon-perturbative" M2-brane Lagrangian comprises the sum of the Nambu–Goto and Wess–Zumino terms, each exhibiting multilinear dependence on the spacetime coordinates.
The dimensional reduction from the M5-brane to the M2-brane is implemented via
the ``non-perturbative projection" $ \partial^m = \epsilon^{ij} \partial_j x^m \partial_i, $
in \bref{stwvMix} and the gauge fixing of the world-volume metric in \bref{gaugechoicewv}.
- Page 13, line 8: The following explanation is added.
This M5-brane extends over both the main space (i.e., the duality-covariant space) and the internal space. Four of its world-volume directions lie in the main space while the remaining directions lie in the internal space, specifically one world-volume direction in the Hamiltonian formalism, or two in the Lagrangian formalism. Considering the critical string action in the full spacetime structure is an interesting subject, although it lies beyond the scope of the present discussion. The relationship between the main space and the internal space is schematically illustrated in Figure 2 the ``slug diagram" (see page 27 of arXiv:1610.0162 or page 14 of \cite{Hatsuda:2023dwx}). In the case of D = 3 the main space coordinate is represented by a bispinor $X^{\alpha\beta}$, and the world-volume coordinate by an antisymmetric bispinor $\sigma^{[\alpha\beta]}$, with $\alpha = 1, \dots, 4$. The internal space coordinate is given by a bispinor $Y^{[\alpha'\beta']}$, where $\alpha' = 1, \dots, 8$. The total number of supersymmetries is 32, which corresponds to the product of the dimensions of the spinor indices $32 = 4 \times 8$.
It is noted that the assignment of the duality symmetric space in ${\cal A}$-theory differs from that in conventional formulations. In ${\cal A}$-theory, the duality-symmetric space is assigned to the main "spacetime" rather than the internal space, such that all tensor gauge fields are automatically incorporated into the coset parameter of $\mathrm{E}_{\mathrm{D+1}} / H$.
- We have corrected typos and grammatical errors as much as possible.

---

## Editorial Decision

published